# A steep switching WSe$_2$ impact ionization field-effect transistor

Haeju Choi[1], Jinshu Li[1], Taeho Kang[1], Chanwoo Kang[1], Hyeonje Son[1], Jongwook Jeon[2], Euyheon Hwang [1,3] ✉ & Sungjoo Lee [1,3] ✉

The Fermi-Dirac distribution of carriers and the drift-diffusion mode of transport represent two fundamental barriers towards the reduction of the subthreshold slope (SS) and the optimization of the energy consumption of field-effect transistors. In this study, we report the realization of steep-slope impact ionization field-effect transistors (I$^2$FETs) based on a gate-controlled homogeneous WSe$_2$ lateral junction. The devices showed average SS down to 2.73 mV/dec over three decades of source-drain current and an on/off ratio of ~10$^6$ at room temperature and low bias voltages (<1 V). We determined that the lucky-drift mechanism of carriers is valid in WSe$_2$, allowing our I$^2$FETs to have high impact ionization coefficients and low SS at room temperature. Moreover, we fabricated a logic inverter based on a WSe$_2$ I$^2$FET and a MoS$_2$ FET, exhibiting an inverter gain of 73 and almost ideal noise margin for high- and low-logic states. Our results provide a promising approach for developing functional devices as front runners for energy-efficient electronic device technology.

After decades of complementary metal-oxide-semiconductor (CMOS) device scaling, the intended improvements in performance and density have been largely achieved. And CMOS devices serve as the main building blocks for modern information processing devices. However, in the past several years, despite advances in crucial processing technologies and the resultant ability to produce smaller feature sizes, it has become clear that conventional device technologies have not kept pace and that scaled device performance has been compromised[1]. Furthermore, as CMOS devices are scaled from generation to generation, power dissipation increases proportionately to increasing transistor density and switching speeds, and power consumption is now becoming a major design challenge[2], particularly for future energy-efficient low-power devices. The low-power operation of metal-oxide-semiconductor field-effect transistors is limited by an inability to scale the subthreshold slope (SS) below 60 mV/dec, which is defined by the Boltzmann distribution of carriers and the drift-diffusion mode of transport in devices. As a consequence of this non-scalability of the SS, additional scaling of

the supply voltage and threshold voltage is prohibited, and the leakage current increases exponentially.

Many efforts have been made to modify the SS in recent years, including altering the carrier injection mode from diffusion to tunneling-based mechanisms[3–6] or mechanical switching operations[7,8]. One such mechanism for overcoming the aforementioned limitation is utilizing impact ionization[9–11]. The impact ionization FET (I$^2$FET) is a reverse-biased p-n junction with a gate-modulated breakdown voltage. The I$^2$FET operation is based on impact ionization triggered by a sufficient electric field, providing the high gain switching property of avalanche breakdown with a sharp current transition from the off to the on state based on the impact ionization phenomenon. Therefore, an SS significantly lower than 60 mV/dec can be achieved. Early-stage experimental research has been conducted using conventional semiconducting materials such as Si[12–14], Ge[15,16], and III–V compounds[10,17]. Although SS values lower than 60 mV/dec have been demonstrated for various fabricated I$^2$FET devices, stringent operational conditions, and practically unacceptable levels of scalabilities for the threshold and

[1]SKKU Advanced Institute of Nanotechnology (SAINT), Sungkyunkwan University, Suwon 440-746, Korea. [2]Department of Electrical and Electronics Engineering, Konkuk University, Seoul 05029, Korea. [3]Department of Nano Engineering, Sungkyunkwan University, Suwon 440-746, Korea. ✉e-mail: euyheon@skku.edu; leesj@skku.edu

supply voltages have been observed based on the limits of fundamental material properties. To overcome this issue, alternative device schemes have been proposed[9,11,13]. However, most previous studies have been limited to theoretical analysis using device simulations because of complicated device structures.

Recent advances in 2D materials and their heterostructures have prompted investigations of steep switching FETs. Such studies are promising and based on the unique properties of 2D materials, various versatile electronic properties[18,19], and the processability of integration on arbitrary substrates without lattice mismatch issues, which facilitates the formation of functional heterostructures[20,21]. Several studies have reported the realization of steep switching transistors fabricated with 2D heterostructures based on quantum mechanical tunneling[3,5,6] and negative capacitance (NC)[22–25] effects. Although experimental realizations of an SS <60 mV/dec have been reported, well-known fundamental tradeoffs for both approaches are still unresolved, including limited on-current drivability for tunneling FETs and hysteresis in NC-FETs. However, very few studies have been reported on I²FETs. One pioneering work was reported recently by Gao et al.[9], who demonstrated a steep SS (0.25 mV/dec) by fabricating a vertical InSe/BP heterostructured bottom-gate I²FET. Their devices also exhibited a low avalanche threshold voltage (<1 V) and low noise figure. Ballistic avalanche phenomena have been claimed as a key point for vertical 2D heterostructure devices. This is likely why device operation with a low SS has been limited to a temperature of 200 K, and room temperature performance has not been demonstrated.

In this paper, we report the realization of a low SS at room temperature by using a homogeneous WSe₂ lateral junction top-gated I²FET. Our devices exhibit average SS values (2.73 mV/dec) through six orders of current at room temperature with a high on/off current ratio of ~10⁶. It is also demonstrated that the threshold and supply voltages in our devices can be scaled down to <0.5 and <0.9 V, respectively. These excellent performance features can be attributed to the high-impact ionization coefficients of WSe₂, which has a relatively short ionization mean free path compared to other layered systems. Because the lucky-drift mechanism of carriers is valid in WSe₂, WSe₂ I²FETs have high impact ionization coefficients and a low SS at room temperature.

Furthermore, a 2D-material-based logic inverter was constructed with a serial connection for the WSe₂ I²FET as a pull-down transistor and MoS₂ FET as a pull-up transistor. The inverters exhibited a high gain of 73, outperforming most of the previously reported 2D-material-based inverters. The noise margins of the inverter, namely $NM_L = 0.506\,V_{DD}$ and $NM_H = 0.493$, were obtained. These values are close to the ideal noise margin ($0.5\,V_{DD}$), suggesting that the WSe₂ I²FET inverter is highly robust to electrical noise from the environment and desirable for integration into multi-stage logic circuits based on its steep switching capacity. Our results provide a promising approach to efficient carrier multiplication through impact ionization and fundamental strategies for finding potential applications in energy-efficient data-centric computing devices.

## Results and discussions

### Fabrication and working principle of WSe₂ I²FET

Figure 1a presents a schematic of the WSe₂ lateral impact ionization transistor. Our device consists of a few-layer WSe₂ structure for the channel, Au electrodes for the source and drain, and an Au local top-gate electrode. We used deposited SiO₂ as a top-gate dielectric to prevent the source- and top-gate electrodes from sticking together and control the charge distribution in the gate-covered region (gated region) evenly using the top gate. The detailed fabrication process is described in Supplementary Note 1a. In this structure, we define the biased electrode connected to the ungated region as the drain and the grounded electrode connected to the gated region as the source. We modulate the conductance of the gated region by applying the top-

gate voltage. In our device, a back-gate voltage is applied to maintain the gated-channel regions of WSe₂ in their intrinsic state (i.e., charge neutral point of the transfer characteristic, see Supplementary Note 3a). Therefore, under weak top-gate voltages (or no gate voltage), the entire region of WSe₂ is intrinsic (we define the effective channel as the intrinsic region). As the top gate voltage increases, the doping level of the gated region increases. When the gated region is fully metalized at a sufficiently high top-gate voltage, the effective channel length is confined to the ungated region, which increases the electric field in that region. The effective channel length can be reduced by reducing the length of the ungated region. Based on the reduction in the effective channel length, the strength of the electric field increases. As a result, carriers gain sufficiently high energy to excite electron-hole pairs through impact ionization, and carrier multiplication occurs in this reduced ungated region.

When the system is partially metalized, the current-voltage characteristics can be understood through the schematic energy band diagrams of the device, as shown in Fig. 1. The gated region is fully p-doped under a negative top-gate voltage (i.e., $V_{GS} < V_{TH}$, where $V_{TH}$ is the threshold voltage for impact ionization), whereas the ungated region is intrinsic. Therefore, a p⁺-i abrupt homojunction is formed at the boundary between the gated and ungated regions. In addition, the potential difference is built up at the boundary and the built-in potential is given by $eV_{bi} = E_g/2 + E_{Fh}$, where $E_g$ is the bandgap of a channel material and $E_{Fh}$ is the Fermi energy of highly doped hole gas and is determined by carrier density ($\rho$), i.e., $E_{Fh} = \hbar^2 \pi \rho / m_p$, where $m_p$ is the effective hole mass. When a reverse bias is applied ($V_{DS} < 0$), the built-in potential at the abrupt boundary (p⁺-i junction) blocks the electron current. Although carrier multiplication occurs in the depletion region with a high reverse voltage, the built-in potential barrier blocks effective electronic current. Under a forward bias ($V_{DS} > 0$), holes are injected from the drain electrode, and the hole current begins to flow. Under a moderate bias ($0 < V_{DS} < V_{BR}$, where $V_{BR}$ is the breakdown voltage, i.e., the threshold drain voltage at which the initiation of breakdown occurs), the electric field is not sufficiently strong to produce impact ionization. Consequently, the current is limited by the reverse current of the p⁺-i diode. In this case, the drain current remains the same as the saturation current. As the drain voltage increases further (i.e., the voltage becomes higher than $V_{BR}$), the current rises sharply with the drain bias and breakdown occurs. The fundamental mechanism responsible for this breakdown is impact ionization. Under this condition ($V_{DS} > V_{BR}$), the applied large electric field causes carriers to acquire sufficient energy to produce electron-hole pairs through impact ionization. Therefore, the drain current increases sharply for $V_{DS} > V_{BR}$, as illustrated in Fig. 2a.

In the proposed device structure, $V_{BR}$ can be reduced to a value smaller than the well-known fundamental limit for impact ionization[26] (i.e., $V_{BR} < E_g/e$). The partially gated region plays a critical role in reducing the drain voltage required for carrier multiplication. Without the gate voltage (i.e., off-state), the drain-source voltage ($V_{DS}$) produces an electric field, $\varepsilon = V_{DS}/L$, in the channel (see Case 4 in Fig. 1), where $L$ is the entire channel length. If the field is relatively weak, the carriers do not gain sufficient energy for impact ionization. When the top-gate voltage was applied (i.e., on-state), the electric field distribution and potential profile changed, as shown in Case 5 in Fig. 1. The channel is divided into a high-field avalanche (ungated) region ($L_1 < x < L$) and a carrier drift (gated) region ($0 < x < L_1$). The multiplication processes only occur in the avalanche region. As shown in Fig. 1, regardless of the top-gate voltage, the overall voltage drop between the drain and source electrodes is always given by the Fermi level difference between the two metal electrodes, $eV_{DS} = E_{FS} - E_{FD}$, where $E_{FS}$ and $E_{FD}$ are the fermi levels of source and drain electrodes, respectively. We note that the potential difference between the two metal electrodes is given by $\int_0^L \varepsilon dx = V_{DS}$, where $\varepsilon$ is the internal electric field in the channel. However, because of the band bending arising

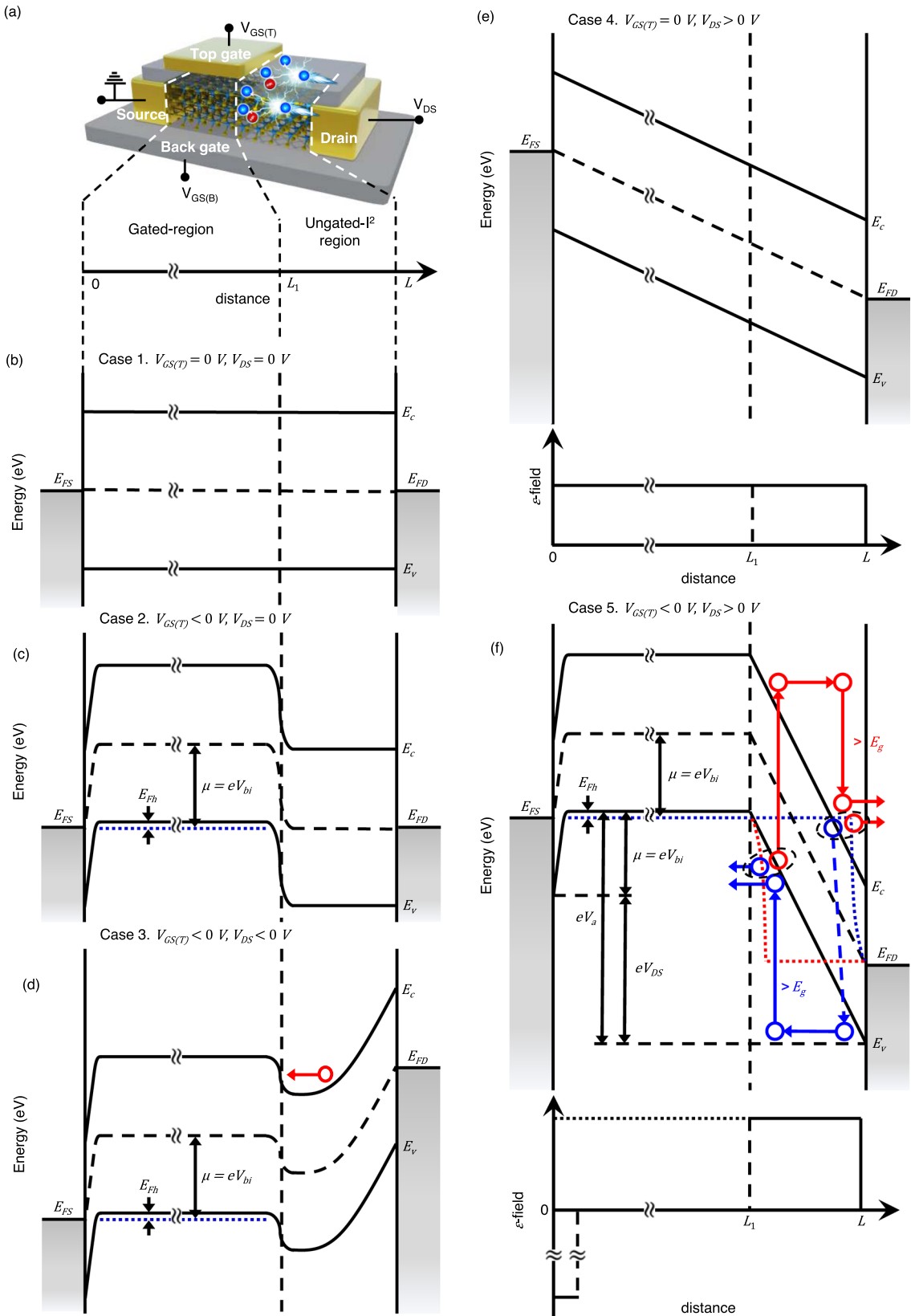

**Fig. 1 | Device structure and energy band profile of the WSe₂ I²FET. a** Schematic of the WSe₂ lateral impact ionization transistor. Energy band diagrams for each case: **b** $V_{GS(T)} = 0\,V$, $V_{DS} = 0\,V$, **c** $V_{GS(T)} < 0\,V$, $V_{DS} = 0\,V$, **d** $V_{GS(T)} < 0\,V$, $V_{DS} < 0\,V$, **e** $V_{GS(T)} = 0\,V$, $V_{DS} > 0\,V$, and **f** $V_{GS(T)} < 0\,V$, $V_{DS} > 0\,V$. Red and blue circles represent electrons and holes, respectively. $V_{GS(T)}$ and $V_{GS(T)}$: top gate and back-gate voltage, $I_{DS}$ and $V_{DS}$: drain current and voltage. $E_C$, $E_V$, $E_{FS}$, and $E_{FD}$: conduction band energy, valence band energy, source fermi level, and drain fermi level, respectively, $E_{FH}$: Fermi energy of highly doped hole gas, $\mu$: chemical potential energy, $E_g$: energy bandgap, $V_a$: potential energy drop in the avalanche region, $L$: total length of the channel, $L_1$: boundary between the gated region and the ungated region, and $V_{bi}$: built-in potential at the boundary between the gated and ungated regions).

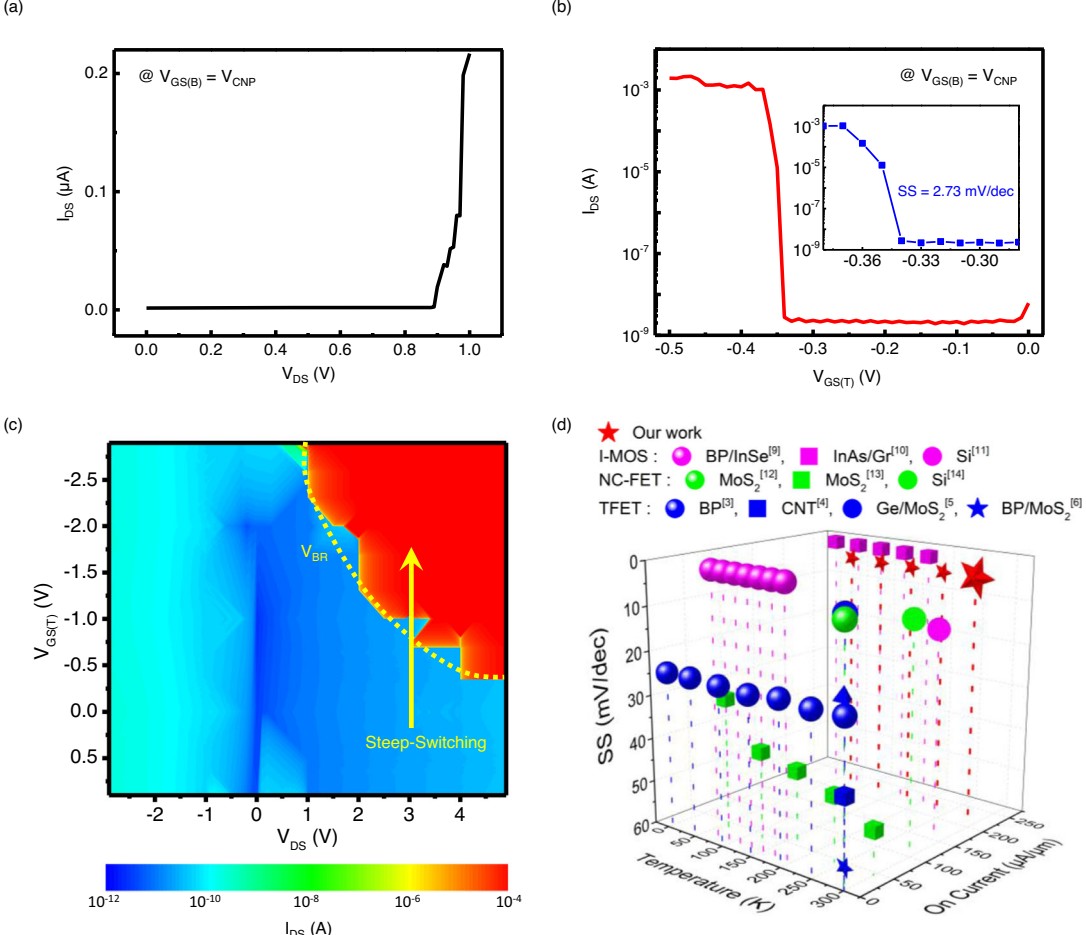

**Fig. 2 | Performances of WSe₂ I²FET. a** $I_{DS}$–$V_{DS}$ characteristics at the charge neutral point ($V_{CNP}$). One can see an abrupt rise over 0.88 V. **b** $I_{DS}$–$V_{GS(T)}$ characteristics at room temperature and a magnified view of the subthreshold region. The on/off current ratio exceeds $10^6$, and the SS is 2.73 mV/dec. **c** Contour plot representing the channel current ($I_{DS}$) as a function of various $V_{DS}$ (from −3 to 5 V) and $V_{GS(T)}$ (from 1 to −3 V). Steep-switching behavior is observed only at the avalanche bias ($V_{DS}$ >0.88 V). The yellow dotted line represents the breakdown voltage ($V_{BR}$) that varies with $V_{GS(T)}$ and $V_{DS}$. **d** Comparison of SS$_{>3\ order}$ (three-order average SS) and $I_{ON}$ (on current) values versus temperature between our work and other steep switching devices (I-MOS: purple, TFET: blue, NC-FET: green). Our device exhibits the lowest SS and highest $I_{ON}$ at room temperature.

from the chemical potential shift in the gated region, the potential energy drop in the avalanche region ($L_1 < x < L$) is given by $eV_a = e\int_{L_1}^{L}\varepsilon dx = eV_{DS} + eV_{bi}$, where $eV_{bi}$ is the built-in potential arising from band bending (see Case 2 in Fig. 1). Because the channel material is set to be intrinsic via the back-gate voltage, the change in chemical potential of the channel is measured from the middle of the bandgap. To achieve energy conservation, a tunneling barrier at the source electrode was placed to compensate for the potential energy drop in the avalanche region, i.e., $e\int_{0}^{L_1}\varepsilon dx = -\mu$. However, this barrier is unrelated to carrier multiplication and may result in an increase in contact resistance. When the top-gated region is fully degenerated, the chemical potential shift is given by $\mu = eV_{bi} = E_g/2 + E_{Fh}$. Thus, depending on the chemical potential shift, the potential energy drop in the avalanche region can be larger than the bandgap, although the applied drain voltage is lower than the bandgap, that is, $eV_a = eV_{DS} + \mu > E_g$. This physical mechanism explains how breakdown (carrier multiplication) occurs even at a drain voltage that is lower than the theoretical limit $V_{DS} < E_g/e$. In addition to the potential difference between the source and drain electrodes, the carriers obtain additional energy for multiplication through band bending. Thus, in our proposed homojunction FET device, the breakdown voltage $V_{BR}$ can be further reduced by simply increasing $\mu$, which is controlled by the top-gate voltage (i.e., $V_{BR} < (E_g - \mu)/e$). In this device structure, since opposite polarity of biases is used for gate and drain terminals, further

scaling of $V_{GS}$ and $V_{DS}$ is required. This issue can be studied further by enhancing gate capacitive coupling using a thin high-κ dielectric and/or by physical reduction of the ungated channel length.

The steep switching results measured via impact ionization are compiled in Fig. 2c as color plots of the channel current ($I_{DS}$) as a function of the applied drain ($V_{DS}$) and gate ($V_{GS(T)}$) voltages. The figure shows the threshold gate voltage that changes according to the applied drain voltage above $V_{BR}$, which is determined by the ungated region length, whereas the current remains within the saturation current region for an applied bias below $V_{BR}$.

For a fixed drain voltage higher than the breakdown voltage ($V_{DS} > V_{BR}$), this steep switching with a small modulation of the top-gate voltage can be understood as an abrupt increase in the electric field caused by a reduction of the effective channel length from the entire channel to the ungated region. At low top-gate voltages ($V_{GS(T)} < V_{TH}$), the entire WSe₂ is intrinsic and no metallic region (i.e., p⁺ layer) appears. Therefore, the entire intrinsic WSe₂ structure becomes the effective channel length. Under this condition, the electric field is below the critical electric field ($E_{CR}$) for impact ionization, and the carriers do not gain sufficient energy to produce impact ionization. As the top-gate voltage increases further, the gated region is metalized. Therefore, the effective channel length of the device decreases, and the electric field of the intrinsic region increases. When the top-gate voltage is higher than $V_{TH}$, the gated region turns into the conduction

channel, and the electric field is concentrated in the ungated region. As a result of this increased electric field, carriers gain sufficient energy to trigger impact ionization, and the number of carriers increases exponentially through avalanche multiplication. Therefore, as shown in Fig. 2b, a very low subthreshold swing of approximately 2.73 mV dec⁻¹ can be obtained by applying a horizontal electric field ($V_{DS} > V_{BR}$) along the short ungated region. In Fig. 2d, we compare the three-order average subthreshold swing ($SS_{>3\ order}$) of our device to other steep switching devices presented in the literature. This figure presents $SS_{>3\ order}$ as a function of temperature and on current. While most devices presented in the literature do not operate at room temperature or exhibit a relatively high SS at room temperature, our device has a low $SS_{>3\ order}$ (~2.73 mV dec⁻¹) and high on/off (>10⁵) ratio for a wide range of temperatures, even while maintaining a low SS at room temperature, which is critical for real device applications.

## Electrical characteristics of WSe₂ I²FET

To investigate the impact ionization phenomenon of WSe₂ in detail, we fabricated various WSe₂ I²FETs with different ungated-region lengths and measured their steep switching transitions under various conditions. The fabrication process and device structure are shown in Supplementary Figs. 1 and 2. Figure 3a presents the transfer curve of the WSe₂ I²FET (gated- and ungated-region lengths of 3 μm and 300 nm, respectively), which exhibits ambipolar transport characteristics with a steep switching transition via impact ionization. When biases in opposite directions are applied to $V_{GS(T)}$ and $V_{DS}$ to sharply bend the band in the ungated region, an applied electric field larger than $E_{CR}$ initiates impact ionization. Figure 3b shows the hole current $I_{DS}$ as a function of drain voltage $V_{DS}$ at a fixed top-gate voltage of $V_{GS(T)} = -1V$. The black (blue) line indicates the measured current on a linear (semilogarithmic) scale. At reverse and low drain voltages, the current increased slightly with drain voltage. However, as the voltage increased further, an abrupt increase in the current was observed, and a steep transition occurred at the breakdown voltage $V_{BR}$, which was attributed to the impact ionization process. We calculated the multiplication factor, defined as $M = \frac{I}{I_{sat}}$[27], where $I_{sat}$ is the saturation current at the $V_{BR}$. The multiplication factor extracted from the measured $I_{DS}$ is presented in Fig. 3c as a function of the electric field ($E = V_{DS}/L$, where $L$ is the effective channel length). A large multiplication factor of up to 10⁶ was observed before permanent breakdown occurred, confirming that the impact ionization process generated a large number of carriers. The breakdown voltage strongly depends on the length of the ungated region of WSe₂. The dependence of the breakdown voltage on the length of the ungated region is shown in Fig. 3d. We used the same conditions for all devices to obtain the length dependence of the WSe₂ I²FETs. After dividing the large WSe₂ flake via etching, it was fabricated such that the gated region length was the same for each device, and only the ungated region length was different. In addition, all measurements were performed using the same $V_{GS(T)}$ of −1 V. The breakdown voltage decreases linearly with ungated region length, which indicates that the critical electric field corresponding to the breakdown voltage is independent of channel length. Therefore, it is expected that further scaling of $V_{DS}$ and $V_{GS}$ is possible using a scaled ungated length of less than 10 nm. The thickness dependence of the critical electric field was also obtained and is shown in Fig. 3e. The field strength increased as the thickness decreased, and it was approximately related to the bandgaps of the samples. In Fig. 3f, $I_{DS}$ normalized by the saturation current, is presented as a function of the electric field for various temperatures at $V_{GS(T)} = -1V$. Steep switching transition via impact ionization can be observed even at room temperature. Overall, the critical electric field for breakdown increases slightly with temperature. Because the temperature dependence of phonon scattering is responsible for changes in the impact ionization coefficients as the temperature changes, the critical electric field increases as the temperature increases. The critical electric fields are typically

observed to be ~300 kV/cm for Si- and Ge-impact ionization-based devices[28,29]. Therefore, the measured $E_{CR}$ for the WSe₂ I²FETs (~70 kV/cm) is relatively small compared to that of other impact ionization transistors, which is attributed to the low threshold energy for ionization and the long inelastic scattering time compared to the ionization mean free path. A lower threshold voltage is expected in materials with lower bandgaps and equal effective masses of electrons and holes. Above the critical electric field, breakdown stems from carrier multiplication through impact ionization. As shown in Fig. 1, carriers gain sufficient energy to produce electron-hole pairs through impact ionization. Therefore, materials with a large bandgap require more energy to trigger impact ionization. A similar investigation of WSe₂ impact ionization properties was performed in the case of a single WSe₂ channel (i.e., all gated regions without ungated regions). Detailed results and discussions are provided in Supplementary Note 2b.

## Theoretical study of impact ionization characteristics

The increase in electron energy depends on the relationship between the acceleration of carriers in the external field and energy dissipation through collision with phonons. Impact ionization also requires the potential for acceleration, meaning there is a minimum width for the space charge region[30]. If the width is greater than the mean free path between two ionizing impacts, then charge multiplication occurs, which can cause an electrical breakdown. If the mean free path is longer than the energy relaxation length and space charge region, then the ionization coefficients in semiconductors can be calculated based on the lucky-drift mechanism[31], where a simple analytical expression was derived and good agreement was obtained for our WSe₂ devices. The lucky-drift mechanism is valid when the momentum relaxation rate is much smaller than the energy relaxation rate ($\tau_m < \tau_E$), which indicates that carriers are accelerated by an external electric field and acquire sufficient energy to produce electron-hole pairs through impact ionization, but no significant energy loss occurs. In our theoretical consideration, there are two main processes in the lucky-drift mechanism, namely the lucky-ballistic mode, where the carrier reaches the threshold without a momentum-relaxing collision, and the lucky-drift mode, where the carrier reaches the threshold without an energy-relaxing collision. The scattering rates by phonons play a critical role in determining the impact ionization coefficients. To understand the breakdown behavior of our samples, we calculated the ionization coefficients of bulk WSe₂ based on the lucky-drift mechanism considering phonon scattering (see Supplementary Note 2d). The calculated ionization coefficients ($\alpha$) of bulk WSe₂ are presented as a function of $E_I/e\xi\lambda$, where $E_I$ is the ionization energy, $\xi$ is the strength of the applied electric field, and $\lambda$ is the mean free path. The total ionization coefficient (solid curve) increases dramatically with an increasing electric field $\xi$ and the lucky-drift process (red curve) is dominant over the lucky-ballistic process (blue curve) for the impact ionization of bulk WSe₂. The phonon energy is principally responsible for changes in the impact ionization coefficients because of temperature changes. With the calculated phonon scattering rate and material parameters of the WSe₂, we determined that $E_I/e\xi\lambda = 4$ - 6. Within this range, the calculated impact ionization coefficients exhibit a slight temperature dependence, as shown in Fig. 4b. Therefore, the breakdown present in the transfer curve in Fig. 4c is relatively insensitive to changes in temperature.

## Complementary inverter fabricated with WSe₂ I²FET

Furthermore, we fabricated a 2D-material-based complementary inverter consisting of an n-type MoS₂ driver transistor and p-type WSe₂ I²FET pull-up transistor and investigated its logic inverting performance. Figure 5b presents the circuit configuration of the MoS₂/WSe₂ inverter. Figure 5c presents the transfer characteristics of the MoS₂ FET and WSe₂ I²FET. The channel length and width of these transistors are

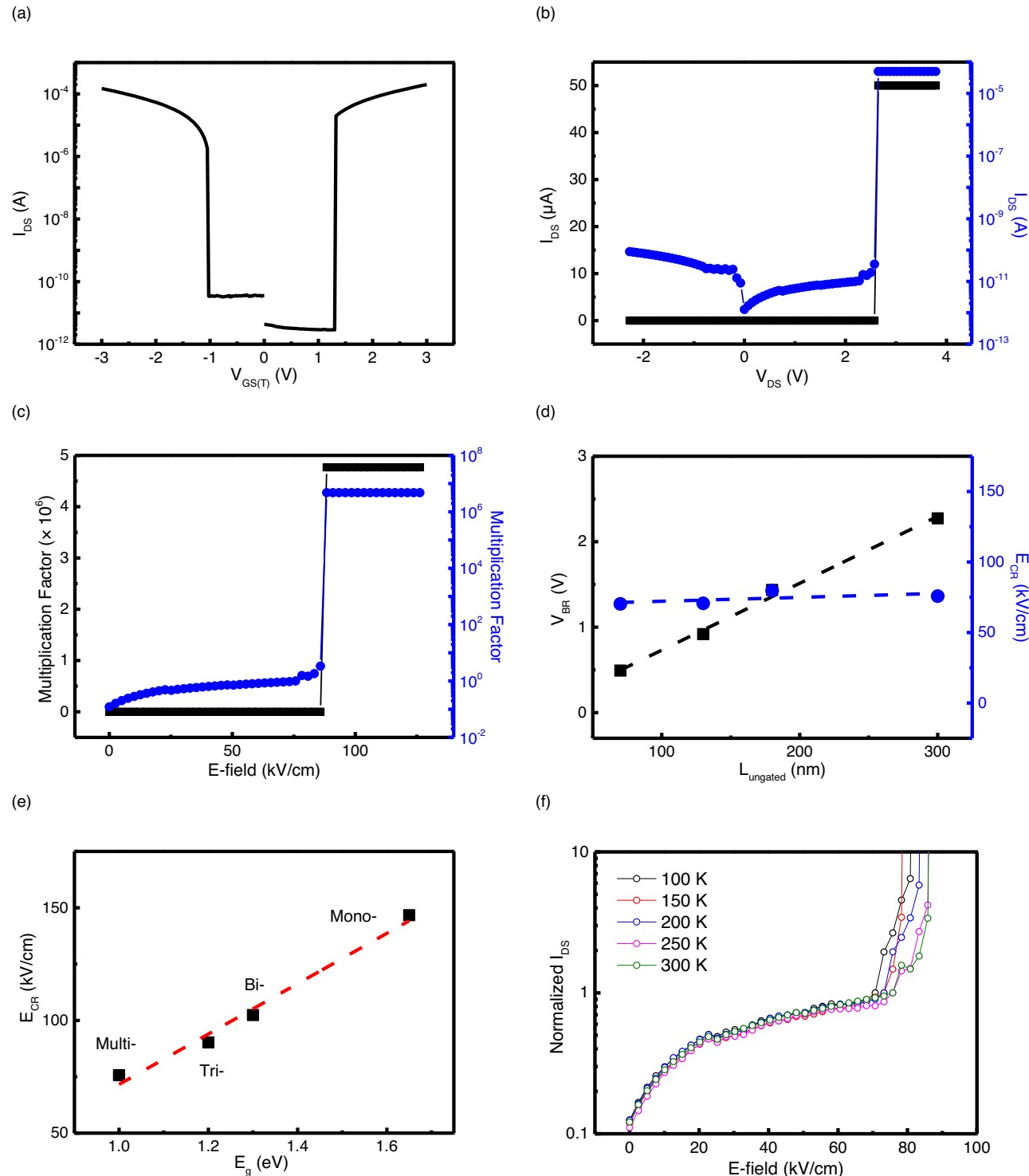

**Fig. 3 | Electrical characteristics of WSe$_2$ I²FET. a** Transfer curve and **b** output curve exhibiting steep switching transition via impact ionization. $I_{DS}$ saturates in the low-E-field region owing to the insulating ungated region, whereas it increases abruptly in the high-E-field region (shown by a log scale and linear scale plotted using blue and black lines, respectively). **c** Calculated multiplication factor (*M*) as a function of electric field. **d** Calculated $V_{BR}$ and $E_{CR}$ of WSe$_2$ I²FETs vs. various ungated channel lengths. ($V_{BR}$: $V_{DS}$ value at which impact ionization is initiated, $E_{CR}$: critical electric field capable of generating impact ionization.). **e** $E_{CR}$ values extracted from various I²FETs with different energy bandgaps vs. number of layers. **f** Measured transfer characteristics as a function of electric field for various temperatures.

2 μm and 5 μm, respectively. The WSe$_2$ I²FET exhibits sharp switching because of hole-induced impact ionization with an SS of approximately 10 mV/dec and on/off ratio of ~10$^5$, whereas the MoS$_2$ FET exhibits typical electron-dominant n-type switching characteristics. A complementary push-pull mode inverter operation was obtained through the in-series connection of these transistors. For a high $V_{IN}$, the n-type MoS$_2$ transistor pulls down $V_{OUT}$ while the p-type WSe$_2$ I²FET acts as the load, and for a low $V_{IN}$, the p-type WSe$_2$ I²FET drives $V_{OUT}$ while the n-type MoS$_2$ transistor acts as the load. Excellent voltage transfer characteristics were obtained from the inverter (Fig. 5d), which can be

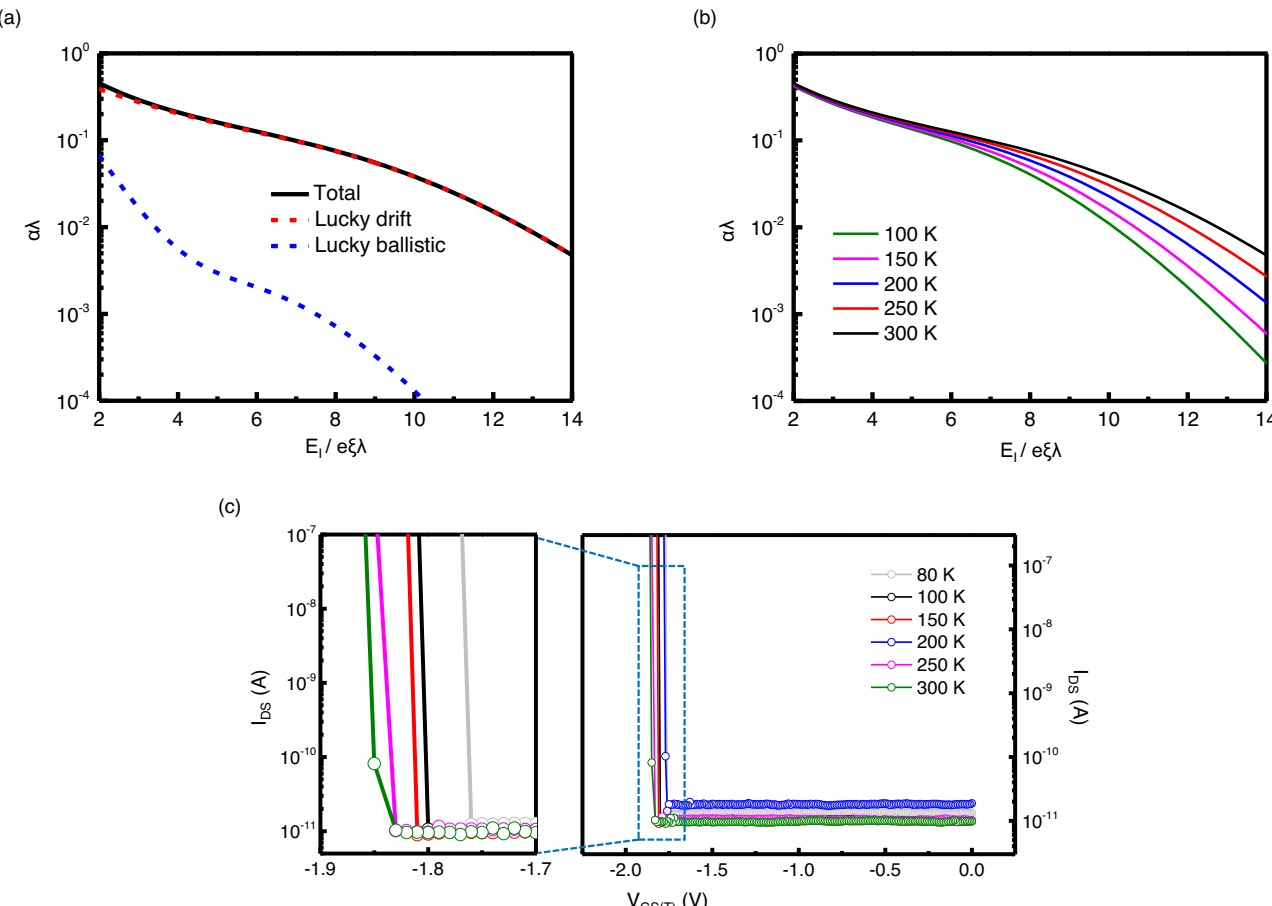

**Fig. 4 | Theoretical clarification of impact ionization. a** Calculated total ionization coefficients of bulk WSe$_2$ at room temperature (solid black curve) as a function of $E_I/e\xi\lambda$, where $E_I$ is the ionization energy, $\xi$ is the strength of the applied electric field, and $\lambda$ is the mean free path. The contributions of the lucky-ballistic mode (red curves) and lucky-drift mode (blue curves) are also shown. **b** Ionization coefficients of the bulk WSe$_2$ at various temperatures. **c** Measured transfer characteristics as a function of top-gate voltage for various temperatures. A magnification view of a blue dashed rectangle is inserted on the left to check the temperature-dependent change of the threshold top-gate voltage. This shows that the WSe$_2$ I$^2$FET is insensitive to temperature.

attributed to the steep drain current transition of the WSe$_2$ I$^2$FET around the DC operation region (green dot in Fig. 5c). The switching threshold voltage of the inverter (green dot in Fig. 5d) corresponds to the operating point in Fig. 5c based on the symmetric balanced driving strengths of both the pull-up and pull-down transistors. This result also suggests that with respect to various demands in different applications, the switching threshold voltage can be further adjusted through the geometrical and/or parameter modulation of each 2D transistor. A high inverter gain ($dV_{OUT}/dV_{IN}$) of -73 was obtained. The approximately ideal sharp and narrow transition in the voltage transfer characteristic yields an excellent noise margin ($NM_L$, $NM_H$ of -50% of $V_{DD}$; see Supplementary Note 4a), which is very desirable for securing low sensitivity to noise and disturbances in real circuit applications. Supplementary Table 2 presents a comparative benchmark of the inverter gain, noise margin, and SS, showing that our inverter outperforms most of the recently reported 2D-material-based steep switching devices and Si devices.

In conclusion, we fabricated an I$^2$FET with a gated region-controlled homogeneous WSe$_2$ junction. The main operating mechanism in our device is the interplay between the on/off switching controlled by the lateral bias (electric field) and impact ionization controlled by the top gate. The applied top-gate voltage in the proposed device controls the carrier multiplication and band energy mismatches, which amplify the on state and suppress the off-state depending on the bottom-gate voltage. The proposed device has an SS much lower than kT/e≈25meV per decade at room temperature. We

reported a low SS (2.73 mV/dec) at room temperature and a high on/off ratio exceeding 10$^6$. These values for the proposed WSe$_2$ I$^2$FETs outperform previously reported steep switching transistors. We theoretically determined that the lucky-drift mechanism of carriers is valid in WSe$_2$, which allows the WSe$_2$ I$^2$FET to have high impact ionization coefficients and a low SS at room temperature. Furthermore, we fabricated a logic inverter with a serial connection of the WSe$_2$ I$^2$FET as a pull-down transistor and a MoS$_2$ FET as a pull-up transistor, demonstrating an excellent inverter gain of 73 and an approximately ideal noise margin for both high- and low-logic states. Our results provide a promising general approach for developing functional devices as front runners for future energy-efficient data-centric computing device technology.

## Methods
### Device fabrication
Multilayer WSe$_2$ flakes (2D semiconductors) were exfoliated on silicon substrates covered with 285 nm of silicon dioxide. The thickness of the WSe$_2$ was first observed using an optical microscope and then verified through atomic force microscopy. For further thickness control or channel definition, inductively coupled plasma was used to treat the WSe$_2$ flakes. Electron-beam lithography and electron-beam evaporation were repeated thrice for each device to fabricate source/drain electrodes (Au 25 nm), a dielectric layer (SiO$_2$ 20 nm), and top-gate electrode (Au 50 nm). The ungated impact ionization region length was controlled by aligning the top-gate electrode from 70 to 420 nm.

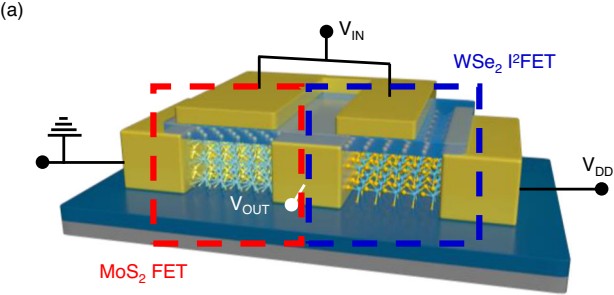

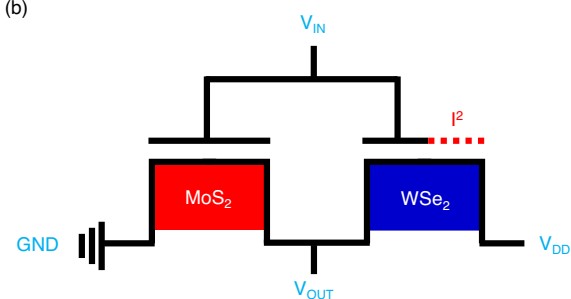

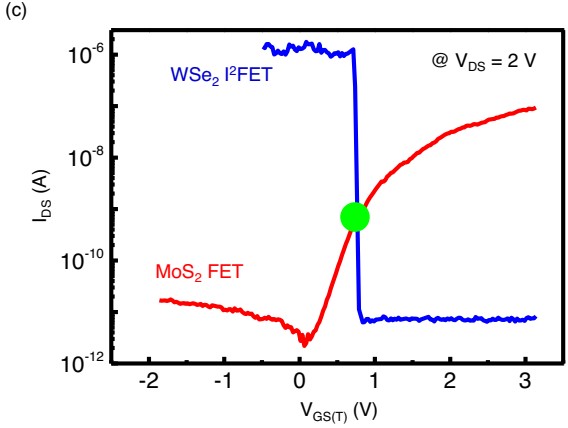

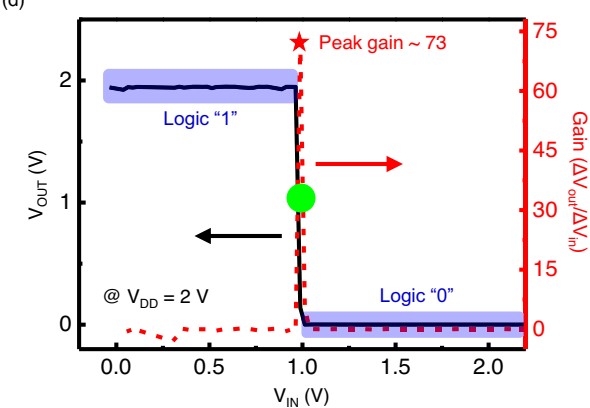

**Fig. 5 | Complementary inverter fabricated with WSe₂ I²FET. a** Schematic of a complementary inverter consisting of an n-type MoS₂ driver transistor and p-type WSe₂ I²FET pull-up transistor. (*V*ᵢₙ and *V*ₒᵤₜ: input and output voltages shared by the two transistors, *V*DD: power supply voltage) **b** Circuit configuration of the inverter. **c** *I*DS–*V*GS(T) curves of the WSe₂ I²FET and MoS₂ transistors (blue and red lines, respectively). **d** Inverter characteristics based on the WSe₂ I²FET in series with n-MoS₂.

See Supplementary Note 1a for a detailed description of the fabrication process.

## Characterization

An optical microscope (Olympus, BX51M) and field-emission scanning electron microscope (JEOL, JSM7500F) were used to confirm the size and shape of the exfoliated WSe₂ flakes and the fabricated devices. The thickness of the flakes and top view of the devices were determined using an atomic force microscope (Park Systems Corp., NX-10) in non-contact mode with PPP-NCHR probe tips (Nanosensors). The cross-sections of the devices were investigated using transmission electron microscopy (JEOL, TEM2100F) at an accelerating voltage of 200 kV. All the electrical properties of the WSe₂ devices were measured using a Keithley 4200 parameter analyzer. Variable temperature measurements were carried out using a hot chuck controller (MS Tech, MST1000H) and cryostat system (MS Tech, VX7).

## Data availability

Relevant data supporting the key findings of this study are available within the article and the Supplementary Information file. All raw data generated during the current study are available from the corresponding authors upon request.

## Code availability

No custom code or mathematical algorithm is used in this study.

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

## Acknowledgements

This research was supported by the Next-generation Intelligence Semiconductor Program and by the Basic Science Research Program through the National Research Foundation of Korea (NRF) funded by the Ministry of Science and ICT (grant numbers: 2020M3F3A2A03082047, 2022M3F3A2A01072215, 2022R1A2C3003068, 2020R1A4A2002806, 2021R1A2C1012176). This work was supported by Samsung Electronics Co., Ltd (IO201215-08197-01).

## Author contributions

S.L. conceived the concept, designed all the experiments, and supervised the work. H.C and J.L. carried out most of the experimental work and data analysis. J.J. and T.K. assisted with the data analysis with all other authors. C.K. and H.S. performed the device fabrication and analysis. E.H. led and supervised the whole revision process, including designing experiments, reorganizing the rationale, and editing the manuscript. All authors discussed the progress of the research and contributed to editing the manuscript.

## Competing interests

The authors declare no competing interests.

## Inclusion & Ethics

All researchers who fulfill the authorship criteria are included as co-authors.
