## [Peer Review File · Nature Communications]

Reviewer comments, first round review –

Reviewer #1 (Remarks to the Author):

The manuscript by Choi et al reports a study on TMDC based impact ionization field-effect transistor (IMOS). They managed to use a partially gating structure to fold a short channel IMOS, where a relatively small V_{ds} is able to induce large band bending of the un-gated channel. Due to giant internal impact ionization gain, the transistor can be operated with ultra-low sub-swing rate (SS). The authors also carefully characterized the avalanche mechanism to support their claim. Overall, the manuscript presents a significant process in post-CMOS electronic devices, including room-temperature low SS at low V_{ds} and up to high current density. I would support its publication after addressing a few questions.

Technically, my major concern is about the robustness or the reproducibility of the device. Historically, IMOS was proposed in silicon early 2000s and unfortunately “abandoned” owing to two challenges. Firstly, a large V_{ds} is generally required to induce avalanche, which prevent cascade of multiple devices. And secondly, hot carriers generated from impact ionization may damage the dielectrics, resulting in very limited device life-time. While the former one is decently solved, the later one is not discussed. How many cycle times could the device work? I saw the authors mentioned cycle-to-cycle variation in the SI. In my opinion, an in-depth discussion is necessary.

And there are also a few minor points.

1. The transfer curves and output curved are given in 2 (a) and (b), but only with very rare curves. Maybe detailed information (more curves with different V_{ds} and V_g) should be given. Or equally, demonstrating more curves with finer V_{ds} steps in Fig 1 d.
2. And also, in my opinion, the way of presentation in Fig 1 d is not clear. A contour plot may be better.
3. Is there any relationship between the gain (multiplication factors M) and channel length? That may help to evaluate the ultimate device scale.

Reviewer #2 (Remarks to the Author):

This work successfully demonstrates an i-MOS with 2D WSe₂ channel, which contributes an useful data to steep-slope device community. However, the well-known fundamental limit of i-MOS is its non-scalable V_{ds} that has to be high enough to trigger impact ionization. It does not make sense for a steep-slope device to be able to scale V_g only. Unfortunately, I do not see any efforts in this work to address this major issue. Thus, the insignificance of this work does not warrant its publication in Nature Comm.

Reviewer #3 (Remarks to the Author):

The manuscript " A steep switching WSe₂ impact ionization field effect transistor" by Haeju Choi and co-workers describe the fabrication of a gated region-controlled homogeneous WSe₂ junction to realize an energy efficient electronic device— I^2 FET. The authors observe a considerable low SS at room temperature with a small bias voltage. Based on the sharp transition of the transfer curve, the authors fabricate an inverter with a high gain of 73.

The topic of energy efficient 2D electronic device is very interesting and, from a technological point of view, the fact that the realization of I^2 FET with low supply voltage and high on/off ratio at room temperature is very impressive. Although previous works have reported that I^2 FET realized a low SS with a small supply voltage at low temperature (Nat. Nanotechnol. 14, 217–222 (2019), Small

Struct. 2, 2000039 (2021) and ACS Nano 14, 434–441 (2020)), it is still interesting to realize these high performances at room temperature. High on/off ratio, low SS at small bias voltage at room temperature have been realized in feedback FET (Electron Devices Meeting (IEDM), 2010 IEEE International. IEEE, 2010, 2008 IEEE International Electron Devices Meeting, 2008, pp. 1-4; Solid State Electronics, Volume 76, p. 109-111). Therefore, it is very important to demonstrate the working mechanism is impact ionization in this paper rather than others. Nonetheless, the data is weakly supported the impact ionization mechanism. The author did not use the gated region-controlled device structure in Fig.1 but a FET to demonstrate the main mechanism of steep switching, which is not convincing. Also, it is quite surprising that the impact ionization can occur at a such small bias voltage, less than E_g/e (E_g is bandgap and e is electron). The reviewer is highly skeptical that the proposed mechanism for this behavior is the result of impact ionization. Therefore, I would not recommend this version of the paper for publication in Nature Communication. These and other critiques are listed below:

1. It is quite interesting to observe the impact ionization with a small threshold voltage $<1V$. But given that the band gap of WSe₂ is large than 1eV, I don't understand how the impact ionization can occur with such a small bias voltage. Even without any energy dissipation, the electron can only gain 1eV of energy from the external bias, which is not enough to excite an electron-hole pairs (Solid-State Electronics Vol. 33, No. 6, pp. 705-718, 1990). The author should give more discussion about how it occurs.
2. The author claim that the on/off ratio exceeds 10^6 with a $<1V$ bias voltage. Which is not consistence with avalanche model. The on/off ration of 10^6 means that the impact ionization will occur at least 20 times, each time it needs a bias of E_g/e (V). Therefore, a bias of $>20V$ is required to realize impact ionization.
3. The author should use a same device structure as in Fig.1 to investigate the mechanism, rather than the different device structure in Fig.2. The low bias voltage, step switch and high on/off ratio are base the device in Fig.1, which has a notable p-n junction. While, the device used to investigate the mechanism shows large threshold voltage of $\sim 15V$ and soft turn-on characteristics. It is not convincing to claim that they have the same mechanism.
4. In order to maintain the impact ionization, there is always a voltage drop across the impact ionization region which is a disadvantage of using I²FET as an inverter. In Fig. 4, the voltage drop across the WSe₂ I²FET device is negligibly small. The author should explain more about this.
5. How to rule out the step transition is due to the mechanism of feedback (2008 IEEE International Electron Devices Meeting, 2008, pp. 1-4; Solid State Electronics, Volume 76, p. 109-111) which shows similar behavior and device structure.

Dear Reviewers:

We appreciate your careful evaluation of our manuscript entitled “**A steep switching WSe₂ impact ionization field-effect transistor**” (Manuscript ID: NCOMMS-22-03294-T) by Choi et al., intended for publication in *Nature Communications*.

Your comments and suggestions have been constructive and helped us improve the quality of the manuscript. After substantial revisions, incorporating all the reviewers’ comments and concerns, we wish to resubmit the manuscript. All concerns and recommendations have been addressed and incorporated in the revised manuscript. Herein, we have responded point-by-point to the reviewers’ comments and identified how and where we have revised the text and figures in the manuscript. We hope that you find the revised manuscript suitable for publication.

We believe that a point-by-point response to the reviewers’ comments and the revised version with new experiments and analysis/discussions provide the reviewers (and readers) with a better understanding of our devices and the significance of our results.

Sincerely,

Euyheon Hwang and Sungjoo Lee

SAINT (SKKU Advanced Institute of Nanotechnology), Sungkyunkwan University (SKKU)

Response to Reviewer 1

We are grateful for this review and appreciate your insightful comments.

Please find below our responses (in blue) to the comments (in black) provided by the reviewers. In addition, revisions to the original article are indicated in red.

Comments:

The manuscript by Choi et al reports a study on TMDC based impact ionization field-effect transistor (IMOS). They managed to use a partially gating structure to fold a short channel IMOS, where a relatively small V_{ds} is able to induce large band bending of the un-gated channel. Due to giant internal impact ionization gain, the transistor can be operated with ultra-low sub-swing rate (SS). The authors also carefully characterized the avalanche mechanism to support their claim. Overall, the manuscript presents a significant process in post-CMOS electronic devices, including room-temperature low SS at low V_{ds} and up to high current density. I would support its publication after addressing a few questions.

Technically, my major concern is about the robustness or the reproducibility of the device. Historically, IMOS was proposed in silicon early 2000s and unfortunately “abandoned” owing to two challenges. Firstly, a large V_{ds} is generally required to induce avalanche, which prevent cascade of multiple devices. And secondly, hot carriers generated from impact ionization may damage the dielectrics, resulting in very limited device life-time. While the former one is decently solved, the later one is not discussed. How many cycle times could the device work? I saw the authors mentioned cycle-to-cycle variation in the SI. In my opinion, an in-depth discussion is necessary.

→ As the reviewer mentioned, robustness or reproducibility is one of the most significant challenges for our device. For a Si-based IMOS, there is a critical limitation in reliability due to damage inflicted when hot carriers generated by impact ionization are injected into the dielectric. However, we believe that this issue can be addressed by our devices owing to (1) greatly reduced scaled V_{ds} (further scaling is possible with a more reduced ungated region length) and (2) highly asymmetric transport properties of 2D materials making carriers flow parallel to the plane, resulting in shifting the main current path away from the dielectric/channel interface. Detailed explanations for each are as follows.

(1) As shown in Fig. R1, the previously reported impact ionization materials require a high drain voltage to generate impact ionization due to the high critical electric field (E_{CR} , usually over 300 kV/cm), which is determined by the intrinsic characteristics of the material and device structure. However, we found that WSe_2 has a low E_{CR} of 20–50 kV/cm, which is a significant finding, indicating that considerable reductions in drain voltage can trigger impact ionization in our device. We need to apply a voltage of 0.6–1.5 V for our WSe_2 I²FET to generate impact ionization in the ungated intrinsic region (i.e., effective channel where impact ionization occurs) of ~ 300 nm, which can be further reduced by using current industry level lithography and patterning methods. With this scaled bias condition, the energy and number of hot carriers will be greatly reduced, and therefore reliability concerns can be addressed.

Figure R1. Comparison of impact ionization coefficient for holes versus field strength at room temperature

(2) In conventional bulk materials used in IMOS, such as Si, a major current flow occurs near the interface in contact with the dielectric, where charges are created by gate-voltage modulation. However, in two-dimensional layered materials such as WSe_2 , the location of the “HOT-SPOT,” where the current mainly flows, is determined by the gate voltage modulation and number of layers [S8]. For a two-dimensional layered system with multiple layers, as the gate voltage increases from the threshold voltage, the HOT-SPOT is located further away from the dielectric. It was reported that when 20 nm SiO_2 , similar to ours, was used as the gate dielectric for 13 layers of the 2D layered material, the position of the HOT-SPOT shifted from 8 to 10.5 layers while $V_{GS} - V_{TH}$ was equal to 0 to 5 V. With a similar analysis applied to our device, it is

expected that the position of the HOT-SPOT changes approximately 20–24 layers below the dielectric with respect to the $V_{GS(T)} - V_{TH}$ change where the transition occurs, as shown in Fig. R2b. This can be significant in suppressing the gate leakage current and hot-carrier-induced damage to the dielectric.

Figure R2. Structure and current flow patterns in (a) conventional I-MOS and (b) our WSe_2 I²FET.

A significant level of gate current generally indicates the poor reliability of the gate dielectric. In particular, the gate leakage current due to injected hot carriers travelling all the way through the dielectric induces critical damage. In the case of Si-based IMOS, the injection efficiency, defined as the ratio of the gate current to the drain current (I_{GS}/I_{DS}), was found to be as high as 10^{-3} for IMOS [S7]. However, our WSe_2 I²FET shows a low gate leakage current of less than 0.1 nA with a low injection efficiency of less than 10^{-5} , as can be seen in Figure S11.

Figure S11. Drain current and gate leakage current versus top-gate voltages of WSe_2 I²FETs with (a) $L_{ungated} = 180$ nm and (b) $L_{ungated} = 420$ nm with $V_{DS} = 3$ V. The gate oxide leakage current has an insignificant value compared with the drain current. (c) Calculated injection efficiency for various devices with different $L_{ungated}$ s compared with conventional IMOS.

The above arguments are incorporated into the revised Supporting Information as follows.

Supplementary Section 3. Properties and control of the WSe₂ I²FET

d. Gate leakage current

Figures S11 (a) and (b) compare the drain and gate leakage currents of WSe₂ I²FETs with different channel lengths, respectively. A significant level of gate leakage current generally indicates poor reliability because although a certain number of carriers make it through the barrier, there are others that get trapped in the oxide or create interface states, causing damage [S7]. Figure S11 (c) exhibits the injection efficiency, defined as the ratio of the gate current to the drain current (I_{GS}/I_{DS}). In contrast to the conventional I-MOS, our device shows a significantly lower injection efficiency, indicating a low hot-carrier trapping rate. Our WSe₂ I²FET shows a low gate leakage current (<0.1 nA with a low injection efficiency $<10^{-5}$).

→ Regarding the reproducibility of the fabricated I²FET, we performed iterative measurements once again under the same conditions for tens of cycles (> 50 cycles), and the results showing repeatability and the analysis through cumulative probability have been added to the revised Supporting Information (Figure. S12) along with the following discussion:

Figure S12. (a) Repeated measurement results recorded every 10 cycles. (b) V_{TH} (black dots) and E_{CR} (blue dots) during repeated steep switching phenomena are shown as cumulative probability curves.

Supplementary Section 3. Properties and control of the WSe₂ I²FET

e. Reliability of the impact ionization phenomenon

Figure S12 (b) illustrates the cumulative probabilities of the threshold voltage (V_{TH}) and critical electric field (E_{CR}) obtained from cycle-to-cycle data (results from the same device over several cycles) over 50 cycles, as shown in Fig. S12 (a). After tens of cycles, similar impact ionization characteristics were observed, indicating that no apparent degradation occurred except for slight changes in the breakdown voltage and current multiplication. Reliability was quantified using the coefficient of variation (CV). CV is commonly used to measure the dispersion of probability distributions and can be calculated using the following expression: $CV = (\sigma/\mu) \times 100$ (%), where σ is the standard deviation, and μ is the absolute mean value.

We believe that the repeatable operation of the fabricated impact-ionization FET can be attributed to factors such as the greatly reduced V_{ds} and highly asymmetric transport properties of 2D materials, making carriers flow parallel to the plane.

First, the previously studied materials usually require a high critical electric field (E_{CR} , usually over 300 kV/cm) to trigger impact ionization. By contrast, we found that WSe₂ has a low E_{CR} of 20–50 kV/cm, which is a very meaningful finding, indicating that a greatly reduced drain voltage can trigger impact ionization in our device. We need to apply a voltage of 0.6–1.5 V for our WSe₂ I²FET to generate impact ionization in the ungated intrinsic region. With this scaled bias condition, the energy and number of hot carriers will be greatly reduced; therefore, the hot-carrier-induced reliability concern can be addressed.

Second, in conventional bulk materials used in IMOS, such as Si, a major current flow occurs near the interface in contact with the dielectric, where charges are created by gate-voltage modulation. However, in two-dimensional layered materials such as WSe₂, the location of the “HOT-SPOT,” where the current mainly flows, is determined by the gate voltage modulation and number of layers [S8]. For a two-dimensional layered system with multiple layers, as the gate voltage increases from the threshold voltage, the HOT-SPOT is located further away from the dielectric. It is expected that the position of the HOT-SPOT changes approximately 20–24 layers below the dielectric, which can be significant in suppressing the gate leakage current (as can be seen in Figure S11) and hot-carrier-induced damage to the dielectric.

References

[S7] Mayer F, Le Royer C, Blachier D, Clavelier L, Deleonibus S. Avalanche breakdown due to 3-D effects in the impact-ionization MOS (I-MOS) on SOI: reliability issues. *IEEE Trans. Electron Devices*. 55, 1373-1378 (2008).

[S8] Das S, Appenzeller J. Where does the current flow in two-dimensional layered systems? *Nano Lett*. 13, 3396-3402 (2013).

And there are also a few minor points.

1. The transfer curves and output curves are given in 2 (a) and (b), but only with very rare curves. Maybe detailed information (more curves with different V_{ds} and V_g) should be given. Or equally, demonstrating more curves with finer V_{ds} steps in Fig 1 d.

→ In accordance with the reviewer's comment, we have measured the transfer and output characteristics under different V_{ds} and V_g conditions, as shown in Figure R3. Typical ambipolar transport characteristics were observed in the transfer curves under various V_{ds} conditions. I_{DS} as a function of the drain voltage V_{DS} at various back gates shows carrier multiplication via impact ionization with a changed charge neutral voltage with respect to V_{GS} .

Figure R3. (a) Transfer curves with different V_{DS} from 1 V to 15 V in increments of 2 V. (b) Output curves with different $V_{GS(B)}$ from -15 V to -30 V with a step of -5 V.

Figure R3b was added to the revised Supporting Information as was Figure S4a.

Figure 1d, with more curves and finer V_{ds} steps, is expressed as a contour plot, as suggested by the reviewer in Question 2 below.

2. And also, in my opinion, the way of presentation in Fig 1 d is not clear. A contour plot may be better.

→ Incorporating the reviewer’s suggestion, Figure 1d has been redrawn with a contour plot, as shown below. The steep-switching behavior (yellow arrow) across the breakdown voltage (yellow dotted line) can be understood by considering the transitions between blue (off state) and red (on state). The relevant explanations in the revised manuscript are as follows:

Figure 1. (d) Contour plot representing the channel current (I_{DS}) as a function of various V_{DS} (from -3 to 5 V) and $V_{GS(T)}$ (from 1 to -3 V).

“The steep-switching results measured via impact ionization are compiled in Fig. 1d as color plots of the channel current (I_{DS}) as a function of the applied drain (V_{DS}) and gate ($V_{GS(T)}$) voltages. The figure shows the threshold gate voltage that changes according to the applied drain voltage above V_{BR} , which is determined by the ungated region length, whereas the current remains within the saturation current region for an applied bias below V_{BR} .”

3. Is there any relationship between the gain (multiplication factors M) and channel length? That may help to evaluate the ultimate device scale.

→ If the channel length is larger than the mean free path, the multiplication factor is independent of the channel length (Fig. S7 (b)). However, a higher drain voltage is required in longer-channel devices to produce carrier multiplication (Fig. S7 (a)).

The revised Figure S7 below exhibits the channel-length-dependent impact ionization characteristics. As previously discussed, V_{BR} increases as the channel length increases; thus, E_{CR} maintains a certain value, which is an intrinsic property of the material. As suggested by the reviewer, to investigate the relationship between the multiplication factor (M) and channel length, the M values of several WSe_2 FETs with different channel lengths are plotted against the applied electric field, as shown in Fig. S7 (b). When an electric field larger than E_{CR} is applied, all M values increase with a similar slope, indicating that M is independent of the channel length. Figure S7 and a brief discussion are included in the revised Supporting Information, as follows:

Figure S7. (a) Output curves for various WSe_2 FETs with different channel lengths. (b) Multiplication factor versus applied electric field corresponding to (a).

Supplementary Section 2. Impact ionization properties of WSe_2

e. Relationship between the multiplication factor and channel length

Figure S7 shows the channel length-dependent impact ionization characteristics. As shown in Figure S7 (a), the V_{BR} decreases linearly as the channel length decreases. It is determined

by E_{CR} , which is an intrinsic property of the material, and follows the relationship of $V_{BR} = E_{CR} \times L$, where L is the channel length. To investigate the relationship between the multiplication factor (M) and channel length, the M values of several WSe₂ FETs with different channel lengths were plotted against the applied electric field, as shown in Figure S7 (b). When an electric field larger than E_{CR} is applied, all M values increase with a similar slope, indicating that M is independent of the channel length.

Response to Reviewer 2

We are grateful for this review and appreciate your insightful comments.

Please find below our responses (in blue) to the comments (in black) provided by the reviewers. In addition, revisions in the original article are indicated in red.

Comments:

This work successfully demonstrates an i-MOS with 2D WSe₂ channel, which contributes an useful data to steep-slope device community. However, the well-known fundamental limit of i-MOS is its non-scalable V_{ds} that has to be high enough to trigger impact ionization. It does not make sense for a steep-slope device to be able to scale V_{gs} only. Unfortunately, I do not see any efforts in this work to address this major issue. Thus, the insignificance of this work does not warrant its publication in Nature Comm.

→ In this paper, we present the experimental findings of our devices, and we observed that to produce carrier multiplication, the applied drain voltage V_{DS} was scalable below the well-known fundamental limit. This result may only be possible in our device structure in which the partially gated region plays a critical role in reducing the drain voltage required for carrier multiplication. Our devices have two gates: one (bottom gate) forces the channel to be intrinsic, which provides the condition for maximum impact ionization, and the other (top gate) controls the channel length and provides an additional potential difference to the channel. Even when a small drain voltage is applied, we can obtain the critical electric field for impact ionization by controlling the top-gate voltage. In our devices, $V_{GS(T)}$ plays two important roles: (1) it reduces the channel length, and consequently, the electric field in the channel increases; and (2) owing to the band mismatch at the boundary, the potential difference in the channel is larger than the applied drain voltage (Figure R4 below). Thus, even though a drain voltage smaller than the theoretical limit ($V_{DS}/e < E_g$) is applied to the device, the total voltage in the channel can be larger than the limit. The effective total voltage becomes $\tilde{V}_{DS} = V_{DS} + \frac{E_g}{2e} + \frac{\mu}{e}$, where E_g is the band gap of the material, and μ is the chemical potential shift (doping level) induced by the top-gate voltage (Figure R4 (d)). Owing to this change in the potential difference, carrier multiplication is possible even at a drain voltage below the theoretical limit, $V_{DS}/e \sim E_g$. We noticed that the drain bias for impact ionization can be further reduced by increasing the capacitive coupling from the

top gate (i.e., increasing the chemical potential).

We believe that breaking the fundamental limit for impact ionization and the fabrication of a steep-slope device working at room temperature are valuable and significant, and therefore this work is suitable for publication in *Nature Communications*.

Figure R4. Energy band profile along the applied bias.

The above discussions and modified Figure 1 are added to the revised manuscript as follows.

Figure 1. (a) Device structure of the WSe₂ I²FET and its energy band profile along the applied bias for each case.

“We note that in our device structure V_{DS} can be reduced to a value smaller than the well-known fundamental limit for impact ionization [26] (i.e., $V_{DS} \sim E_g/e$, where E_g is the band gap of a channel material). The partially gated region plays a critical role in reducing the drain voltage required for carrier multiplication. Without the gate voltage (i.e., off-state), the source-drain voltage (V_{DS}) produces an electric field, $E_1 = V_{DS}/L_1$, where L_1 is the entire channel length. If the field is relatively low, the carriers do not gain sufficient energy for impact ionization. When the top gate voltage was applied (i.e., on-state), we observed a significant change in the voltage difference between the two contact points, which increased as $\tilde{V}_{DS} = V_{DS} + \frac{E_g}{2e} + \frac{\mu}{e}$, where μ is the chemical potential shift (doping level) induced by the top gate voltage (see Fig. 1a). With the increase in gate voltage, the channel lengths also decreased from L_1 and L_2 . Thus, the electric field in the channel became $E_2 = \tilde{V}_{DS}/L_2$, which is higher than the field expected without an additional potential drop, V_{DS}/L_2 . This physical mechanism explains how breakdown (carrier multiplication) occurs even at a drain voltage less than the theoretical limit, $V_{DS} \sim E_g/e$. (Page 5)

References

[26] Anderson C, Crowell C. Threshold energies for electron-hole pair production by impact ionization in semiconductors. *Phys. Rev. B.* **5**, 2267 (1972).

Response to Reviewer 3

We are grateful for this review and appreciate your insightful comments.

Please find below our responses (in blue) to the comments (in black) provided by the reviewers.

In addition, revisions to the original article are indicated in red.

Comments:

The manuscript "A steep switching WSe₂ impact ionization field effect transistor" by Haeju Choi and co-workers describe the fabrication of a gated region-controlled homogeneous WSe₂ junction to realize an energy efficient electronic device—I²FET. The authors observe a considerable low SS at room temperature with a small bias voltage. Based on the sharp transition of the transfer curve, the authors fabricate an inverter with a high gain of 73.

The topic of energy efficient 2D electronic device is very interesting and, from a technological point of view, the fact that the realization of I²FET with low supply voltage and high on/off ratio at room temperature is very impressive. Although previous works have reported that I²FET realized a low SS with a small supply voltage at low temperature (Nat. Nanotechnol. 14, 217–222 (2019), Small Struct. 2, 2000039 (2021) and ACS Nano 14, 434–441 (2020)), it is still interesting to realize these high performances at room temperature. High on/ratio, low SS at small bias voltage at room temperature have been realized in feedback FET (Electron Devices Meeting (IEDM), 2010 IEEE International. IEEE, 2010, 2008 IEEE International Electron Devices Meeting, 2008, pp. 1-4; Solid State Electronics, Volume 76, p. 109-111). Therefore, it is very important to demonstrate the working mechanism is impact ionization in this paper rather than others. Nonetheless, the data is weakly supported the impact ionization mechanism. The author did not use the gated region-controlled device structure in Fig.1 but a FET to demonstrate the main mechanism of steep switching, which is not convincing. Also, it is quite surprising that the impact ionization can occur at a such small bias voltage, less than E_g/e (E_g is bandgap and e is electron). The reviewer is highly skeptical that the proposed mechanism for this behavior is the result of impact ionization. Therefore, I would not recommend this version of the paper for publication in Nature Communication. These and other critiques are listed below:

1. It is quite interesting to observe the impact ionization with a small threshold voltage $<1V$. But given that the band gap of WSe₂ is large than 1eV, I don't understand how the impact ionization can occur with such a small bias voltage. Even without any energy dissipation, the electron can only gain 1eV of energy from the external bias, which is not enough to excite an electron-hole pairs (Solid-State Electronics Vol. 33, No. 6, pp. 705-718, 1990). The author should give more discussion about how it occurs.

→ This is a surprising finding, and it may only be possible in our device structure in which the partially gated region plays a critical role in reducing the drain voltage needed for carrier multiplication. Without the gate voltage, that is, the off-state (Figure R4 (c)), an applied source-drain voltage (V_{DS}) produces a potential incline and electric field to the device. Carriers were injected from the drain electrode by overcoming the barrier formed at the contact. However, because of the long channel length (L_1), the electric field (V_{DS}/L_1) is relatively low, and the carriers do not gain sufficient energy for impact ionization. When the top gate voltage was applied, that is, the on-state (Figure R4 (d)), we observed a significant change in the voltage between the two contacts, which increased according to $\tilde{V}_{DS} = V_{DS} + \frac{E_g}{2e} + \frac{\mu}{e}$, where E_g is the bandgap of the material, and μ is the chemical potential (doping level) by the top gate voltage (see Figure R4 (d)). With the increase in gate voltage, the channel lengths also decreased from L_1 and L_2 . Thus, the electric field (\tilde{V}_{DS}/L_2) in the channel was higher than the expected value without an additional potential drop, V_{DS}/L_2 . Owing to this change in the potential difference, carrier multiplication is possible even at a drain voltage below the theoretical limit, $V_{DS}/e \sim E_g$. We noticed that the drain bias for impact ionization can be reduced further by increasing the capacitive coupling from the top gate (i.e., increasing the chemical potential).

Figure R4. Energy band profile along the applied bias.

Revised Figure 1 and corresponding explanations are added to the revised manuscript as follows.

Figure 1. (a) Device structure of the WSe₂ I²FET and its energy band profile along the applied bias for each case.

“We note that in our device structure, V_{DS} can be reduced to a value smaller than the well-known fundamental limit for impact ionization [26] (i.e., $V_{DS} \sim E_g/e$, where E_g is the band gap of a channel material). The partially gated region plays a critical role in reducing the drain voltage required for carrier multiplication. Without the gate voltage (i.e., off-state), the source-drain voltage (V_{DS}) produces an electric field, $E_1 = V_{DS}/L_1$, where L_1 is the entire channel length. If the field is relatively low, the carriers do not gain sufficient energy for impact ionization. When the top gate voltage was applied (i.e., on-state), we observed a significant change in the voltage difference between the two contact points, which increased as $\tilde{V}_{DS} = V_{DS} + \frac{E_g}{2e} + \frac{\mu}{e}$, where μ is the chemical potential shift (doping level) induced by the top gate voltage (see Figure 1(a)). With the increase in gate voltage, the channel lengths also decreased from L_1 and L_2 . Thus, the electric field in the channel became $E_2 = \tilde{V}_{DS}/L_2$, which is higher than the field expected without an additional potential drop, V_{DS}/L_2 . This physical mechanism explains how breakdown (carrier multiplication) occurs even at a drain voltage less than the theoretical limit, $V_{DS} \sim E_g/e$.

(Page 5)

References

[26] Anderson C, Crowell C. Threshold energies for electron-hole pair production by impact ionization in semiconductors. *Phys. Rev. B.* **5**, 2267 (1972).

2. The author claim that the on/off ratio exceeds 10^6 with a $<1V$ bias voltage. Which is not consistence with avalanche model. The on/off ration of 10^6 means that the impact ionization will occur at least 20 times, each time it needs a bias of E_g/e (V). Therefore, a bias of $>20V$ is required to realize impact ionization.

→ The on/off ratio does not represent the number of multiplications in the devices. At the zero top gate voltage ($V_{GS(T)}=0$), carrier injection into the channel is very inefficient because of the wide potential barrier at the contact. However, when the top gate voltage was applied, the potential barrier became narrower; therefore, more carriers could enter the channel (see Figure R5). Thus, the ratio of the number of carriers injected into the channel in the on-current to off-current was proportional to e^{W_1/W_2} , where W_1 and W_2 indicate the average tunneling barrier widths of the off- and on-current situations, respectively.

Therefore, a high bias voltage is not required to obtain a very high on/off ratio.

Figure R5. Energy band diagram with (black and red lines) and without (green dotted line) applied gate voltage

If the on/off ratio is determined solely by carrier multiplication via impact ionization, according to the avalanche model, which expects the proliferation of a solitary carrier owing to avalanche multiplication with $2^{2^{n-1}}$, a high on/off ratio can be achieved in a few multiplications.

3. The author should use a same device structure as in Fig.1 to investigate the mechanism, rather than the different device structure in Fig.2. The low bias voltage, step switch and high on/off ratio are base the device in Fig.1, which has a notable p-n junction. While, the device used to investigate the mechanism shows large threshold voltage of ~15V and soft turn-on characteristics. It is not convincing to claim that they have the same mechanism.

→ We agree with Reviewer 3 regarding this comment. To explain the general impact ionization properties of WSe₂ (not our device) and check the thickness dependence, we used the device structure of Figure 2. Because our devices were fabricated with WSe₂, we believe this figure is necessary. We note that as long as the channel length is larger than the mean free path of the carriers, the multiplication properties of WSe₂ are independent of the channel length scaled in terms of the electric field. Thus, a device with a longer channel length has a higher threshold voltage. In the revised version, we have added details of the working mechanism of the device based on Figure 1. We believe that Figure 2 is helpful for readers to understand the general multiplication properties of WSe₂; therefore,

we have retained this figure in the revised version.

Nevertheless, in accordance with the reviewer's comment, similar investigations of WSe₂ I²FET have been included in the revised manuscript. Although there appears to be a slight variation from the structural difference that enables steep switching, the same tendency of the breakdown voltage to decrease with respect to the length scaling of the region was observed. The calculated E_{CR} value is larger than that of the FET because the voltage distribution between the gated and ungated regions was not considered.

Figure S13 and relevant discussions are included in the revised Supporting Information as follows.

Figure S13. (a) Output curve exhibiting steep-switching transition via impact ionization. (b) Calculated multiplication factor (M) as a function of the electric field. (c) Calculated V_{BR} and E_{CR} of WSe₂ I²FETs with various ungated region lengths.

Supplementary Section 3. Properties and control of the WSe₂ I²FET

f. Analysis of impact ionization characteristics in a WSe₂ I²FET

Because the investigation of the impact ionization characteristics of WSe₂ through I²FET is complicated owing to its gated-region controlled system, we analyzed it through a simple FET structure, as shown in Figure 2, to show the superior impact ionization characteristics of WSe₂. Nevertheless, to prove that the steep switching of our I²FET is based on the mechanism of impact ionization, we performed the same analysis as in Figure 2. Figure S13 shows the impact ionization characteristics demonstrated by WSe₂ I²FET, which are the same as the device in Figure 1. Although there appears to be a slight variation from the structural difference that enables steep switching, the tendency is the same as the breakdown voltage decreases according to the length scaling of the region where impact

ionization occurs. The calculated E_{CR} value is larger than that of the FET because the voltage distribution between the gated and ungated regions was not considered.

4. In order to maintain the impact ionization, there is always a voltage drop across the impact ionization region which is a disadvantage of using I^2FET as an inverter. In Fig. 4, the voltage drop across the WSe_2 I^2FET device is negligibly small. The author should explain more about this.

→ Like the conventional I-MOS, our device has a voltage drop across the ungated impact ionization region, as the reviewer noted. Previously reported I-MOS devices require a high bias for operation owing to their high E_{CR} ; therefore, a high voltage drop occurs when used in an inverter. However, as discussed in Question 1, our device can operate with scaled V_{DS} , indicating that a small voltage drop is expected. Figure S15 (a) shows the VTC curve of an inverter comprising an I^2FET device with an ungated region length of approximately 300 nm. At this time, a voltage drop of approximately 0.7 V, similar to the theoretically expected voltage drop of $20 \text{ kV/cm} \times 300 \text{ nm} = 0.6 \text{ V}$, was experimentally measured. The voltage drops in inverters composed of different I^2FETs with various ungated region lengths are summarized in Fig. S15 (b). The blue dashed line indicates the expected ideal voltage drop calculated from the investigated E_{CR} of 20 kV/cm, and the experimental values indicated by the black squares follow the same trend. As shown in Figure 4, we accomplished a complementary inverter with an enhanced voltage drop by simply reducing the ungated region length to 25 nm.

Figure S15 and relevant discussions are included in the revised Supporting Information as follows.

Figure S15. (a) Inverter characteristics based on the WSe_2 I²FET (with ungated region length of 300 nm) in series with n-MoS₂ FET. (b) Calculated expected voltage drop (blue dashed line) and experimentally measured voltage drops (black squares) with various ungated region lengths.

Supplementary Section 4. Complementary inverter with WSe_2 I²FET

b. Scalable voltage drop of WSe_2 I²FET inverter

Because our WSe_2 I²FET is based on an impact ionization mechanism, there is a voltage drop over the region where the impact ionization occurs. However, this voltage drop follows the relationship of $V_{BR} = E_{CR} \times L$, where L is the channel length, and can be reduced by reducing the channel length. Figure S15 (a) shows the VTC curve of an inverter composed of an I²FET device with an ungated region length of approximately 300 nm. At this time, a voltage drop of approximately 0.7 V, similar to the theoretically expected voltage drop of $20 \text{ kV/cm} \times 300 \text{ nm} = 0.6 \text{ V}$, was experimentally measured. The voltage drops in inverters composed of different I²FETs with various ungated region lengths are summarized in Fig. S15 (b). The blue dashed line indicates the expected ideal voltage drop calculated from the investigated E_{CR} of 20 kV/cm, and the experimental values indicated by the black squares follow the same trend.

5. How to rule out the step transition is due to the mechanism of feedback (2008 IEEE International Electron Devices Meeting, 2008, pp. 1-4; Solid State Electronics, Volume 76, p. 109-111) which shows similar behavior and device structure.

→ In an FB-FET, the current is controlled by potential barriers near the source and drain electrodes. These potential barriers next to the electrodes are a key element in the FB-FET because they return all or part of the output to the input. The amplified carriers flow when the barriers are removed, which results in a high on-off ratio and steep switching in the FB-FET. However, in our device, the potential barriers are not built next to the electrodes, and once the carriers enter the channel, they flow easily. Depending on the intensity of the electric field, carrier multiplication occurs in the channel, which results in a high on-off ratio and steep switching. Thus, the two devices work in a completely different way, and we have added the difference between the two devices in the Discussion section of the revised manuscript.

The feedback field-effect transistor (FB-FET) mentioned by the reviewer is similar to our device, with a partially gated structure. This forms a potential barrier that restricts carrier transport in the channel and operates by controlling this barrier. For the feedback mechanism to occur with steep switching, an energy band of p-n-p-n or p-n-i-p-n must be formed to accumulate carriers in the potential wall near the electrode. This was possible in the references using a p-i-n structure on the channel, but since we only have an i-region in contrast to the FB-FET, it is fundamentally impossible to form such energy bands.

Figure R6. (a) Schematic view of the FB-FET. (b) Output characteristics of the FB-FET showing large I_D - V_D hysteresis. (c) Energy band profile of the FB-FET under constant

$V_G = -2$ V with various V_D . The electron and hole injection barriers are indicated by the double arrows, which block the carriers flow at low V_D and are eliminated at $V_D = -2$ V owing to strong feedback. (Solid State Electronics, Volume 76, p.109-111, 2012)

Furthermore, owing to the aforementioned feedback mechanism, as illustrated in Figure R6 (c), the FB-FET has the unique characteristic of having a large hysteresis, as demonstrated in Figure R6 (b), because it operates when carriers accumulate in the local area of the channel. However, there was very little hysteresis during the abrupt current increase with impact ionization in our WSe₂ FET and I²FET devices (Figure R7 (a) and (b)), suggesting that our devices do not operate with a feedback mechanism.

Figure R7. (a) I_{DS} - V_{DS} output characteristics of WSe₂ FETs. (b) Hysteresis curves of WSe₂ I²FETs. Sweep directions are marked by arrows.

Figure R7 (a) has been added to the revised Supporting Information as was Figure S5.

Figure R7 (b) has been added to the revised Supporting Information as Figure S10, with a brief additional explanation as follows.

“In addition, this little hysteresis supports that our device is based on an impact ionization mechanism rather than a feedback mechanism.”

Reviewer comments, second round review –

Reviewer #1 (Remarks to the Author):

I reviewed the revised manuscript. And I think both of my and the other reviewers' comments are well addressed. I support its publication.

Reviewer #2 (Remarks to the Author):

I am the second reviewer in the first-round review. I appreciate the authors' efforts in improving the band diagram sketch and providing more detailed explanation in the main text, which help reviewers comprehend what the authors meant to present in this manuscript. However, I regret to say these efforts did not address my previous comments: this work does not address the fundamental issue ($V_{ds} > E_g/q$ for triggering impact ionization) of i-MOS based on the device operation mechanism described by the authors in Figure 1. As pointed out by the 3rd reviewer, to trigger impact ionization in semiconductors, carriers have to gain sufficient energy, i.e. $> E_g$. In the revised manuscript, the authors claim that a band bending is introduced by the combination of top and bottom gates to allow the carriers to gain $> E_g$ energy even when V_{ds} is smaller than E_g/q . I'd like to remind the authors that the maximum amount of energy that carriers can gain in a FET device is solely determined by V_{ds} , more specifically, the Fermi level difference between source and drain. Band bending can only tune the local electric field, which might help the impact ionization rate, but is irrelevant with the triggering threshold of impact ionization. I'd suggest the authors to redraw the last plot in Figure. 1a, by adding quasi-Fermi levels for electrons and holes (as well as the energy level of those e-h pair cartoons w.r.t Fermi levels), and making the band bending and bias in scale. I believe an improved plot can help the authors understand my comments above.

Another major concern is that the way to operate this device basically prevents it from scaling supply voltage, regardless of the real mechanism of this device. Please note that the bias directions of this device are the opposite for gate and drain. In such condition, V_{gd} is larger than V_{ds} and V_{gs} . In other words, supply voltage is determined by V_{gd} , instead of the traditional V_{gs} and V_{ds} . Thus, this device, although realized steep slope, does not help save energy. In summary, I do not recommend to accept this manuscript to Nature Comm.

Reviewer #3 (Remarks to the Author):

The manuscript "A steep switching WSe₂ impact ionization field-effect transistor" by Haeju Choi and co-workers now has been revised. But the replies are quite confusing. I hope the authors can be more careful to illustrate the results. I still cannot recommend this revised version to be published in Nature Communications. The critiques are listed below:

1. Why was $E_g/2e$ added to the equation on line 139? If it is true, what is the physical meaning?
2. I agree that the junction barrier can reduce the impact ionization voltage, but it is a minimum value for one impact ionization process. I don't think the impact ionization process occurs only once in the sample.
3. In the second reply, the on/off ratio is not thought of as the number of multiplications. I'm confused, the WSe₂ channel without the top gate area will be very insulating without enough multiplication. How does it transport the carrier? Can the authors elaborate on how electrons/holes are transported through WSe₂?
4. I still think the author should use the same device structure as in Fig.1 to investigate the mechanism, rather than the different device structure in Fig.2. Did the author have any difficulties in making the same device?

Response to Reviewer 1

Comments:

I reviewed the revised manuscript. And I think both of my and the other reviewers' comments are well addressed. I support its publication.

We appreciate the positive and encouraging comments from Reviewer 1.

Response to Reviewer 2

We are grateful for your review and appreciate your comments.

Please find our responses (in blue) to the comments below (in black). In addition, revisions of the original article are indicated in red.

Comments:

I am the second reviewer in the first-round review. I appreciate the authors' efforts in improving the band diagram sketch and providing more detailed explanation in the main text, which help reviewers comprehend what the authors meant to present in this manuscript. However, I regret to say these efforts did not address my previous comments: this work does not address the fundamental issue ($V_{ds} > E_g/q$ for triggering impact ionization) of i-MOS based on the device operation mechanism described by the authors in Figure 1. As pointed out by the 3rd reviewer, to trigger impact ionization in semiconductors, carriers have to gain sufficient energy, i.e. $> E_g$. In the revised manuscript, the authors claim that a band bending is introduced by the combination of top and bottom gates to allow the carriers to gain $> E_g$ energy even when V_{ds} is smaller than E_g/q . I'd like to remind the authors that the maximum amount of energy that carriers can gain in a FET device is solely determined by V_{ds} , more specifically, the Fermi level difference between source and drain. Band bending can only tune the local electric field, which might help the impact ionization rate, but is irrelevant with the triggering threshold of impact ionization. I'd suggest the authors to redraw the last plot in Figure. 1a, by adding quasi-Fermi levels for electrons and holes (as well as the energy level of those e-h pair cartoons w.r.t Fermi levels), and making the band bending and bias in scale. I believe an improved plot can help the authors understand my comments above.

→ We agree that the maximum amount of energy that carriers can gain in a typical FET device is solely determined by V_{DS} . However, unlike a typical FET, in our homojunction FET device, the carriers obtain additional energy for carrier multiplication through band bending. Below, we explain how our devices obtain additional energy and break the fundamental limit of impact ionization (i.e., $eV_{DS} < E_g$). Based on the reviewer's suggestion, in the revised manuscript, we have redrawn Figure 1a in scale. We believe that the revised figures will help readers better understand the physical working mechanism of our device.

To understand the impact of the ionization process under a drain voltage less than the bandgap of the channel material, we show the electric field distribution and potential profile in Fig. R1a. The Fermi level difference between the source and the drain is fixed and given by eV_{DS} . When the top gate is turned off, the device becomes a typical FET structure, and in this case, carriers do not obtain sufficient energy for carrier multiplication because the maximum energy of a carrier is less than the energy gap of the channel (i.e., $V_{DS} < E_g/e$). However, when the top gate voltage was applied, the electric field distribution and potential profile changed, as shown in Fig. R1b. The channel is divided into a high-field avalanche (ungated) region and a carrier drift (gated) region. The multiplication processes only occur in the avalanche region. The voltage drop across the avalanche region is given by $V_a = V_{DS} + \mu/e$, where $\mu = E_g/2 + E_F$ is the chemical potential measured from the middle of the gap of the channel material, and E_F is the Fermi energy, which is determined by the induced carrier density (ρ), i.e., $E_F = \hbar^2\pi\rho/m_p$. Because we set the channel material to be intrinsic via the back-gate voltage (i.e., the chemical potential is located in the middle of the band gap), the change in chemical potential of the channel is measured from the middle of the band gap. When the channel is highly doped, the Fermi energy is below the top of the valence band, and the Fermi energy is given by the free carriers.

Figure R1. Energy band profile along the applied bias (a) without gate bias and (b) with gate bias.

As shown in the figure, the overall voltage drop between the drain and source electrodes is always given by the Fermi level difference between the two metal electrodes eV_{DS} . The potential difference between the two electrodes is given by $\int_0^L \mathcal{E} dx = V_{DS}$, where \mathcal{E} is the internal electric field. However, the potential energy drop in the avalanche region ($L_1 < x < L$) is given by

$$eV_a = e \int_{L_1}^L \mathcal{E} dx = eV_{DS} + \mu > E_g,$$

which can be larger than the bandgap, although the applied drain voltage is less than the bandgap. To satisfy energy conservation, the tunneling barrier at the source electrode was placed to compensate for the potential energy drop in the avalanche region, that is, $e \int_0^{L_1} \mathcal{E} dx = -\mu$. However, this barrier is unrelated to carrier multiplication and can result in an increase in contact resistance.

→ Based on the above discussion and reviewer's suggestion, in the revised manuscript, we have modified the manuscript and added the corresponding figures in Figure 1a as follows.

Figure 1. (a) Device structure of the WSe₂ I²FET and its energy band profile along the applied bias for each case.

“In the proposed device structure, V_{BR} can be reduced to a value smaller than the well-known fundamental limit for impact ionization [26] (i.e., $V_{BR} < E_g/e$, where E_g is the band gap of a channel material). The partially gated region plays a critical role in reducing the drain voltage required for carrier multiplication. Without the gate voltage (i.e., off-state), the drain-source voltage (V_{DS}) produces an electric field, $\mathcal{E} = V_{DS}/L$, in the channel (see Case 4 in Fig. 1a), where L is the entire channel length. If the field is relatively weak, the carriers do not gain sufficient energy for impact ionization. When the top gate voltage was applied (i.e., on-state), the electric field distribution and potential profile changed, as shown in Case 5 in Fig. 1a. The channel is divided into a high-field avalanche (ungated) region ($L_1 < x < L$) and a carrier drift (gated) region ($0 < x < L_1$). The multiplication processes only occur in the avalanche region. As shown in Fig. 1a, regardless of the top gate voltage, the overall voltage drop between the drain and source electrodes is always given by the Fermi level difference between the two metal electrodes, eV_{DS} (i.e., the potential difference between the two metal electrodes is given by $\int_0^L \mathcal{E} dx = V_{DS}$, where \mathcal{E} is the internal electric field in the channel). However, because of the band bending arising from the chemical potential shift in the gated region, the potential energy drop in the avalanche region ($L_1 < x < L$) is given by $eV_a = e \int_{L_1}^L \mathcal{E} dx = eV_{DS} + \mu$, where μ is the chemical potential shift measured from the middle of the bandgap (see Case 2 in Fig. 1a). Because the channel material is set to be intrinsic via the back-gate voltage, the change in chemical potential of the channel is measured from the middle of the bandgap. To achieve energy conservation, a tunneling barrier at the source electrode was placed to compensate for the potential energy drop in the avalanche region, i.e., $e \int_0^{L_1} \mathcal{E} dx = -\mu$. However, this barrier is unrelated to carrier multiplication and may result in an increase in contact resistance. When the top gated region is fully degenerated, the chemical potential shift is given by $\mu = E_g/2 + E_F$, where E_F is the Fermi energy of the free carriers and is determined by carrier density (ρ), i.e., $E_F = \hbar^2 \pi \rho / m_p$, where m_p is the effective hole mass. Thus, depending on the chemical potential shift, the potential energy drop in the avalanche region can be larger than the bandgap, although the applied drain voltage is

lower than the bandgap, that is, $eV_a = eV_{DS} + \mu > E_g$. This physical mechanism explains how breakdown (carrier multiplication) occurs even at a drain voltage that is lower than the theoretical limit $V_{DS} < E_g/e$. In addition to the potential difference between the source and drain electrodes, the carriers obtain additional energy for multiplication through band bending. Thus, in our proposed homojunction FET device, the breakdown voltage V_{BR} can be further reduced by simply increasing μ , which is controlled by the top-gate voltage (i.e., $V_{BR} < (E_g - \mu)/e$.”

(page 5-6)

Another major concern is that the way to operate this device basically prevents it from scaling supply voltage, regardless of the real mechanism of this device. Please note that the bias directions of this device are the opposite for gate and drain. In such condition, V_{gd} is larger than V_{ds} and V_{gs} . In other words, supply voltage is determined by V_{gd} , instead of the traditional V_{gs} and V_{ds} . Thus, this device, although realized steep slope, does not help save energy. In summary, I do not recommend to accept this manuscript to Nature Comm.

→ As pointed out by the reviewer, the operation of our device was demonstrated under the opposite bias directions for V_{GS} and V_{DS} . Our responses to this comment can be explained from two perspectives.

(1) Regarding the energy consumption issue, the power dissipation of the logic gates can be characterized under two modes: static power consumption ($P_{static} = V_{DD} \times I_{supply}$) and dynamic power consumption ($P_{dynamic} = C_L \times V_{DD}^2 \times f$).

In our device structure, the gate dielectric is required to cover the gated region, not the ungated region, that is, capacitive coupling between the gate and drain terminals can be minimized. Therefore, the leakage current and capacitance between the gate and drain terminals are not a major concern for total power consumption. In addition, because the potential levels of the gated region and ungated region edges were identical, the overall gate leakage current was insignificant, as shown in Figure S12.

Regarding the dynamic power consumption ($P_{dynamic}$), V_{DD} does not represent the maximum voltage between terminals (i.e., V_{GD}), but the voltage swing at each node

(i.e., V_{GS} and V_{DS} , respectively) where the actual voltage transition happens. Therefore, dynamic power can be written as: $P_{dynamic} = P_{dyn_gate} + P_{dyn_drain} = C_L \times V_{GS}^2 \times f + C_L \times V_{DS}^2 \times f$. In this regard, we believe that the advantages of the obtained steep switching at room temperature from the proposed device are still retained in terms of energy savings.

- (2) Apart from the above discussion, further scaling of V_{GS} and V_{DS} can be achieved by enhancing capacitive coupling or by physically scaling the ungated region length.

Firstly, the enhancement of gate coupling can result in further scaling of V_{GS} and V_{DS} . In our current device, steep switching with impact ionization was observed under a V_{DS} bias below the bandgap, which was attributed to the changes in the potential profile and electric field distribution. Once the voltage drop across the avalanche region reaches the bandgap, impact ionization is triggered, resulting in steep switching. Controlling the chemical potential (μ) is a key factor in meeting this condition. The chemical potential can be further increased by enhancing the capacitive coupling of the channel from the top gate, which ensures satisfying impact-ionization conditions with scaled V_{GS} and V_{DS} . In the proposed device, ~ 20 nm of SiO_2 was used as the top-gate dielectric material. The chemical potential and energy band diagrams are shown in Fig. 1a. The electric field in the ungated region is higher than the critical electric field of WSe_2 . By employing a thin high- κ dielectric (HfO_2 : 1-5 nm, $\kappa = 25$), μ was estimated to be enhanced by 0.29-0.57 eV. Consequently, the identical potential profile of the experimentally-observed condition in this study (shown by a green dotted line in Figure R2) can be achieved using scaled V_{GS} and V_{DS} , as shown using orange and red dotted lines in Figure R2. If we consider the potential profile for the onset of impact ionization (i.e., $V_a \approx E_g$), V_{GS} and V_{DS} can be further scaled. In the near future, we hope to report new experimental results by utilizing a thin HfO_2 gate dielectric in the WSe_2 channel (currently, the ALD- HfO_2 deposition process is being investigated), which will confirm the possibility of voltage scaling from the device structure.

Figure R2. Contour plot representing channel current (I_{DS}) as a function of V_{DS} and $V_{GS(T)}$.

Secondly, scaling down the length of the ungated region, where impact ionization occurs, can result in further V_{DS} scaling. Our updated experimental results, including steep switching with a scaled-down ungated channel length of 70 nm (shown in Figure R3 and in revised Figure 2d), show that V_{DS} can be scaled down to 0.7 V. We intend to investigate further possible scaling of the V_{DS} bias by fabricating further scaled ungated channels with lengths below 10 nm, something that is not achievable with our current lithography capability.

Figure R3. I_{DS} - V_{DS} characteristics of WSe₂ I²FET using a 70-nm ungated-channel length

Figure 2(d). V_{BR} and E_{CR} of WSe₂ I²FETs using various ungated channel length

Although the focus of our current manuscript is the demonstration of a new conceptual steep-switching transistor that can overcome the fundamental limit, we hope that the bias scaling issue can be addressed through further engineering efforts. This concern has been briefly included in the revised manuscript as follows:

“In this device structure, since opposite polarity of biases is used for gate and drain terminals, further scaling of V_{GS} and V_{DS} is required. This issue can be studied further by enhancing gate capacitive coupling using a thin high- κ dielectric and/or by physical reduction of the ungated channel length.”

(page 6)

Response to Reviewer 3

We are grateful for this review and appreciate your comments.

Please find our responses (in blue) to the comments provided (in black). In addition, revisions of the original article are indicated in red.

Comments:

The manuscript "A steep switching WSe₂ impact ionization field-effect transistor" by Haeju Choi and co-workers now has been revised. But the replies are quite confusing. I hope the authors can be more careful to illustrate the results. I still cannot recommend this revised version to be published in Nature Communications. The critiques are listed below:

1. Why was $E_g/2e$ added to the equation on line 139? If it is true, what is the physical meaning?

→ $E_g/2$ arises from the change in chemical potential when the top gate voltage is applied. It does not have any specific physical meaning, and is simply given because we set the reference point as the middle of the bandgap. Figure R4 shows the schematic of the band structure. When $V_{GS(T)} = 0$, the chemical potential (μ) is located in the middle of the bandgap (we set the system to be intrinsic by applying the back-gate voltage). When a negative top gate voltage is applied, the system becomes hole-doped, and the chemical potential moves down to the valence band; eventually, at a high gate voltage, the chemical potential is below the top of the valence band, which is the condition for impact ionization in our device. In this case, the change in the chemical potential becomes $\mu = E_g/2 + E_F$ when measured from the middle of the bandgap, where E_F is the Fermi energy of free carriers, and is determined solely by the carrier density (ρ) in the valence band, i.e., $E_F = \hbar^2 \pi \rho / m_p$, where m_p is the effective hole mass.

Figure R4. Energy band profile of gated-region (a) without and (b) with applied gate bias.

2. I agree that the junction barrier can reduce the impact ionization voltage, but it is a minimum value for one impact ionization process. I don't think the impact ionization process occurs only once in the sample.

→ We agree with the reviewer that one carrier multiplication occurs only once at the minimum voltage for impact ionization if we consider only one type of carrier. However, the actual multiplication process is more complex because both electrons and holes contribute to the multiplication process. When a carrier (e.g., a hole) enters the multiplication region at minimum voltage for impact ionization, four carriers (two holes and two electrons) exit the region after one cycle of the process (see Figure R5). The current is expressed as $\Delta Q/\Delta t$. Thus, the measured current was used to count the number of carriers exiting the region per given time interval. If we assume that one carrier enters the region every nano-second, then, without the multiplication, the current is measured to be $i_1 \sim 0.1$ nA, which corresponds to the current of the off-state. However, when multiplication occurs, the current becomes $i_2 = 4Ni_1$, where N is the number of multiplications occurring during the given time interval (i.e., one nano-second). Usually, one cycle of the process occurs in a very short time (in the order of ps, which is estimated from $\tau \sim \frac{L}{v_d} = \frac{L}{\mu E_{CR}}$, where v_d , E_{CR} , and μ are the drift velocity, critical electric field, and mobility of a sample, respectively). Thus, if it takes 1 ps to complete one cycle of the

process, the current with multiplication theoretically becomes $i_2 \sim 10^3 i_1$. In a real device, the process time depends on the mobility, and a higher-mobility sample is expected to have a higher on-off ratio.

Figure R5. (a) Schematic illustration of impact ionization process. (b) A simplified process of impact ionization with high electric field. M_n and M_p are the avalanche gain due to electrons and holes, respectively.

3. In the second reply, the on/off ratio is not thought of as the number of multiplications. I'm confused, the WSe₂ channel without the top gate area will be very insulating without enough multiplication. How does it transport the carrier? Can the authors elaborate on how electrons/holes are transported through WSe₂?

→ In Fig. 1a of the manuscript, we explain how the current flows in our device, and we have included the following discussion of the figure in the revised manuscript. Without a top-gate voltage, our device was a standard FET device. Thus, carriers were transported through the channel (seen in the upper panel in Fig. R6) with a finite bias voltage. Even when a top-gate voltage was applied (seen in the lower panel in Fig. R6), the channel (ungated region) remained insulated (semiconductor); the current did not flow without a bias voltage. Applying a top-gate voltage simply reduces the channel length, and the device remains a standard FET at zero gate voltage. The current flows with only a finite bias voltage. Because the drain current is proportional to the bias voltage, a very low current flows at a low bias, and without multiplication. However, when the bias voltage is larger

than the breakdown voltage for impact ionization, many carriers are produced in the ungated region; therefore, high current flows. Because the top gating induces bending and provides additional energy to the carriers, our device has a very low breakdown voltage and a shows steep switching behavior.

Figure R6. Energy band profile of WSe₂ I²FET for each case.

4. I still think the author should use the same device structure as in Fig.1 to investigate the mechanism, rather than the different device structure in Fig.2. Did the author have any difficulties in making the same device?

→ As suggested by the reviewer, in the revised manuscript, Figure 2 was redrawn using the results obtained from a fabricated device with the same structure as that seen in Figure 1 (i.e., WSe₂ I²FET with a partial gated structure). A relevant discussion has been included in the revised manuscript, as follows.

Figure 2. (a) Transfer curve and (b) output curve exhibiting steep-switching transition via impact ionization. I_{DS} saturates in the low-E-field region owing to the insulating ungated region, whereas it increases abruptly in the high-E-field region (shown by a log scale and linear scale plotted using blue and black lines, respectively). (c) Calculated multiplication factor (M) as a function of electric field. (d) Calculated V_{BR} and E_{CR} of WSe₂ I²FETs vs. various ungated channel lengths. (e) E_{CR} values extracted from various I²FETs with different energy bandgaps vs. number of layers. (f) Measured transfer characteristics as a function of electric field for various temperatures.

“To investigate the impact ionization phenomenon of WSe₂ in detail, we fabricated various WSe₂ I²FETs with different ungated-region lengths and measured their steep switching transitions under various conditions. The fabrication process and device structure are

shown in Figures S1 and S2. Figure 2a presents the transfer curve of the WSe₂ I²FET (gated- and ungated-region lengths of 3 μm and 300 nm, respectively), which exhibits ambipolar transport characteristics with a steep switching transition via impact ionization when the potential energy drop in the ungated-region is larger than the bandgap. When biases in opposite directions are applied to $V_{GS(T)}$ and V_{DS} to sharply bend the band in the ungated region, an applied electric field larger than E_{CR} initiates impact ionization. Figure 2b shows the hole current I_{DS} as a function of drain voltage V_{DS} at a fixed top-gate voltage of $V_{GS(T)} = -1$ V. The black (blue) line indicates the measured current on a linear (semilogarithmic) scale. At reverse and low drain voltages, the current increased slightly with drain voltage. However, as the voltage increased further, an abrupt increase in the current was observed, and a steep transition occurred at the breakdown voltage V_{BR} , which was attributed to the impact ionization process. We calculated the multiplication factor, defined as $M = I/I_{sat}$ [27], where I_{sat} is the saturation current at the V_{BR} . The multiplication factor extracted from the measured I_{DS} is presented in Fig. 2c as a function of the electric field ($E = V_{DS}/L$, where L is the effective channel length). A large multiplication factor of up to 10^6 was observed before permanent breakdown occurred, confirming that the impact ionization process generated a large number of carriers. The breakdown voltage strongly depends on the length of the ungated region of WSe₂. The dependence of the breakdown voltage on the length of the ungated region is shown in Fig. 2d. We used the same conditions for all devices to obtain the length dependence of the WSe₂ I²FETs. After dividing the large WSe₂ flake via etching, it was fabricated such that the gated region length was the same for each device, and only the ungated region length was different. In addition, all measurements were performed using the same $V_{GS(T)}$ of -1 V. The breakdown voltage decreases linearly with ungated region length, which indicates that the critical electric field corresponding to the breakdown voltage is independent of channel length. Therefore, it is expected that further scaling of V_{DS} and V_{GS} is possible using a scaled ungated length of less than 10 nm. The thickness dependence of the critical electric field was also obtained and is shown in Fig. 2e. The field strength increased as the thickness decreased, and it was approximately related to the bandgaps of the samples. In Fig. 2f, I_{DS} normalized by the saturation current, is presented as a function of the electric field for various temperatures. Steep-switching transition via impact ionization can be observed even at room temperature. Overall, the critical electric field for breakdown

increases slightly with temperature. Because the temperature dependence of phonon scattering is responsible for changes in the impact ionization coefficients as the temperature changes, the critical electric field increases as the temperature increases. The critical electric fields are typically observed to be ~ 300 kV/cm for Si- and Ge-impact ionization-based devices [28, 29]. Therefore, the measured E_{CR} for the WSe₂ I²FETs (~ 70 kV/cm) is relatively small compared to that of other impact ionization transistors, which is attributed to the low threshold energy for ionization and the long inelastic scattering time compared to the ionization mean free path. A lower threshold voltage is expected in materials with lower bandgaps and equal effective masses of electrons and holes. Above the critical electric field, breakdown stems from carrier multiplication through impact ionization. As shown in Fig.1, carriers gain sufficient energy to produce electron–hole pairs through impact ionization. Therefore, materials with a large bandgap require more energy to trigger impact ionization. A similar investigation of WSe₂ impact ionization properties was performed in the case of a single WSe₂ channel (i.e., all gated regions without ungated regions). Detailed results and discussions are provided in Supplementary Section 2b.”

(page 7-9)

As we believe that Figure 2 in the original manuscript is still helpful for readers to understand the impact ionization properties of WSe₂ material, it has been re-written and moved to the revised Supplementary Information, as follows.

Supplementary section 2. Impact ionization properties of WSe₂

b. Impact ionization characteristics for various lengths, thicknesses, and temperatures

Figure S4. (a) Transfer curve and (b) output curve exhibiting impact ionization in the high- V_{DS} regime ($V_{DS} < -13.8$ V). I_{DS} increases linearly in the low-E-field regime, whereas it increases abruptly in the high-E-field regime (log scale and linear scale plotted with blue and black lines, respectively). The inset shows the schematic of the WSe₂ FET. (c) Calculated multiplication factor (M) as a function of the electric field. (d) Calculated V_{BR} and E_{CR} of WSe₂ FETs with various channel lengths. (e) E_{CR} values and multiplication factors in the same electric field ($E = 52$ kV/cm) with different energy band gaps varying with the number of layers. (f) Normalized I_{DS} as a function of electric field for various temperatures measured at $V_{GS(B)} = -30$ V.

“We fabricated a simple WSe₂ FET in which carrier doping in the channel was modulated

only by the back gate (see the inset in Fig. S4a; the fabrication process is shown in Fig. S1). Figure S4a shows the transfer curve of the WSe₂ FET (channel length: ~4.8 μm), which has typical ambipolar transport characteristics and exhibits hole doping for $V_{GS(B)} < -15$ V and electron doping for $V_{GS(B)} > -15$ V. Figure S4b shows the hole current I_{DS} as a function of drain voltage V_{DS} at a fixed back-gate voltage of $V_{GS(B)} = -25$ V. The black (blue) line indicates the measured current on a linear (semilogarithmic) scale. At low drain voltages, the current increased with drain voltage and approached saturation values at an intermediate voltage. However, as the voltage increased further (i.e., $V_{DS} < -13.8$ V), an abrupt increase in the current was observed, and breakdown occurred at breakdown voltage V_{BR} , which was attributed to the impact ionization process. The multiplication factor (M) extracted from the measured I_{DS} is presented in Fig. S4c as a function of the electric field ($E = V_{DS} / L$, where L is the channel length). A large multiplication factor of up to 5,000 was observed before permanent breakdown occurred, confirming that the impact ionization process generated a large number of carriers. The breakdown voltage depends strongly on the channel length and thickness of WSe₂. The channel length dependence on the breakdown voltage is shown in Fig. S4d. To obtain the length dependence of the multilayer WSe₂ FETs, we used the same conditions for all devices (the large WSe₂ flake was divided into flakes of different lengths via etching). The breakdown voltage increases linearly with channel length, indicating that the critical electric field corresponding to the breakdown voltage is independent of channel length. We obtained an E_{CR} of approximately 30 kV/cm for the multilayer WSe₂ FETs. The thickness dependence of the critical electric field was also obtained, as shown in Fig. S4e. The calculated E_{CR} s were 51.2, 41.2, 37.6, and 31.5 kV/cm for mono-, bi-, tri-, and multi-layer WSe₂, respectively. The field strength increased as the thickness decreased and was approximately related to the bandgaps of the samples. Figure S4e also shows the thickness-dependent multiplication factors (shown with blue dots) measured at the same electric field ($E = 52$ kV/cm), for comparison. In Fig. S4f, I_{DS} normalized by the saturation current is presented as a function of the electric field for various temperatures at $V_{GS(B)} = -30$ V.”

Reviewer comments, third round review –

Reviewer #2 (Remarks to the Author):

I am the second reviewer in the last two round of review.

I appreciate that the authors invested more efforts in improving band diagrams to help readers understand the device operation mechanism the authors intended to present.

The band diagram in this version exposes the authors' misunderstanding of carrier transport in FET device. Electron/hole reservoirs are inside contacts, thus Fermi levels (E_f) stem from source/drain contacts, and defines the V_{ds} , i.e., $V_{ds}=(E_{fd}-E_{fs})/q$. In the band diagram (Figure R1b), the authors marked V_{ds} and Fermi level in a confusing way, making the locations of E_{fs} and E_{nd} unclear. Utilizing the symbols in Figure R1b, E_{fs}/E_{fd} is either at '0eV'/' E_i ' or '-0.5eV'/' E_v ', carrier transport is limited within this 0.88eV energy range. The top gate can only enhance the electric field, but cannot expand this energy range. To be more specific, if E_{fs} is at '0eV', the energy range between ' E_i ' and ' E_v ' does not contribute to carrier transport; if E_{fs} is at '-0.5eV', the energy range between '0eV' and '-0.5eV' does not contribute to carrier transport, and ' E_f ' was marked by the authors in a wrong place. In short, not all the band bending energy range in the ungated region can contribute to avalanche, the definition of ' V_a ' in the diagram is inappropriate.

Regarding my second concern that the gate and drain bias are in the opposite direction, although the authors proposed some knobs the rebuttal to suppress this effect, this fundamental issue is still limiting the energy efficiency improvement margin of this device.

In summary, since the mechanism of this device remains confusing, I do not recommend to accept this paper at this stage.

Reviewer #3 (Remarks to the Author):

I reviewed the revised manuscript. And I think my comments are well addressed. I support its publication.

Response to Reviewer #2

We are grateful for your review and appreciate your comments.

Please find our responses (in blue) to the comments below (in black). In addition, revisions made to the original article are indicated in red.

Comments:

I am the second reviewer in the last two round of review.

I appreciate that the authors invested more efforts in improving band diagrams to help readers understand the device operation mechanism the authors intended to present.

The band diagram in this version exposes the authors' misunderstanding of carrier transport in FET device. Electron/hole reservoirs are inside contacts, thus Fermi levels (E_f) stem from source/drain contacts, and defines the V_{ds} , i.e., $V_{ds}=(E_{fd}-E_{fs})/q$. In the band diagram (Figure R1b), the authors marked V_{ds} and Fermi level in a confusing way, making the locations of E_{fs} and E_{nd} unclear.

Utilizing the symbols in Figure R1b, E_{fs}/E_{fd} is either at '0eV'/' E_i ' or '-0.5eV'/' E_v ', carrier transport is limited within this 0.88eV energy range. The top gate can only enhance the electric field, but cannot expand this energy range. To be more specific, if E_{fs} is at '0eV', the energy range between ' E_i ' and ' E_v ' does not contribute to carrier transport; if E_{fs} is at '-0.5eV', the energy range between '0eV' and '-0.5eV' does not contribute to carrier transport, and ' E_f ' was marked by the authors in a wrong place.

In short, not all the band bending energy range in the ungated region can contribute to avalanche, the definition of ' V_a ' in the diagram is inappropriate

→ We thank Reviewer #2 for suggesting the inclusion of a clear figure to help readers understand the mechanism of device operation. In the previous response, E_f in Figure R1b was not the Fermi energy of the source or drain electrode but represented the Fermi level of the heavily degenerated hole gas arising from the gate voltage. In the revised figure, we have placed all symbols at their correct locations, including the Fermi levels of the source (E_{FS}) and drain (E_{FD}) electrodes and the Fermi energy of highly degenerated holes (E_{Fh}) of the channel material.

We agree with the reviewer about the carrier transport of FETs and we are fully aware of the flow of current in the FETs. However, we want to emphasize that the FET considered in our study contains a highly degenerated region. We point out that the reviewer's

comment of “Utilizing the symbol ... in a wrong place.” is not applicable to our partially gated homojunction device. This comment is correct for ungated FETs. In highly doped FETs, the channel acts as a conducting material such as a metal, and a small voltage can cause current flow. For example, in a highly doped Si-MOFET, current flows even under a very small drain voltage (i.e., microvolts). Thus, the minimum drain voltage required for the current in FETs depends on doping (or gating) conditions.

We now show how our FET structure obtains additional energy to produce impact ionization even when the source–drain voltage is smaller than the fundamental limit, $eV_{DS} < E_g$. Figure R(a) and (b) show the energy-band diagrams at the equilibrium state of a traditional FET device and our FET devices, respectively, where E_{FS} and E_{FD} denote the Fermi levels of the source and drain electrodes, respectively. When a sufficiently high top-gate voltage is applied, the channel is fully degenerated and the Fermi level of holes (E_{Fh}) is below the top of the valence band. The Fermi energy of the degenerate region is determined by the induced carrier density (n) as $E_{Fh} = \frac{\pi\hbar^2}{m}n$. Figure R(b) shows the energy-band diagram of our FET device, wherein the channel is partially gated. When the top-gate voltage is applied to the gated region, the doping level of the gated region increases, whereas the ungated region is in the intrinsic state (i.e., the chemical potential is located at the middle of the band gap). When the gated region is highly degenerated, a p⁺-i abrupt homojunction is formed at the boundary of the gated and ungated regions, as shown in Figure R(b) (right panel). Thus, the potential difference is built up at the boundary and the built-in potential is given by $eV_{bi} = \mu = E_g/2 + E_F$. Figure R(c) and (d) show the energy band diagrams of an ordinary FET and our FET, respectively, with both the top-gate voltage and the applied drain voltage. The Fermi energies of the electrodes and hole gas are indicated by the same symbols that are used in Figure R(a) and (b). The energy band of the gated channel is barely affected by the drain voltage because the highly degenerated region of the channel is fully metalized. Thus, holes injected from the drain electrode feel a potential difference of $V_a = V_{DS} + V_{bi}$ instead of simply V_{DS} , which is the potential difference between the source and drain voltages (i.e., $eV_{DS} = E_{FS} - E_{FD}$). As shown in the figure, the main difference between the ordinary FET and our FET is the potential profile near the contact point. In an ordinary FET, the potential acts as a transport barrier and results in contact resistance. However, in our FET, the whole potential V_a is extended to the ungated region, causing carriers to gain sufficient energy to excite electron–hole pairs

even when the applied voltage is smaller than the band gap $V_{DS} < E_g/e$. All additional energies obtained by band bending do not transfer to carriers, but the main point we want to emphasize is that if carriers can obtain part of the energy from bend banding, breaking the fundamental limit of impact ionization (i.e., $eV_{DS} < E_g$) is possible.

Figure R. Energy band diagrams at the equilibrium state of (a) a traditional FET device and (b) our FET devices. Energy band diagrams of (c) an ordinary FET and (d) our FET with both top-gate and applied drain voltages.

Regarding my second concern that the gate and drain bias are in the opposite direction, although the authors proposed some knobs the rebuttal to suppress this effect, this fundamental issue is still limiting the energy efficiency improvement margin of this device.

In summary, since the mechanism of this device remains confusing, I do not recommend to accept this paper at this stage.

→ Regarding the energy efficiency issue, we responded in the 2nd revision that the proposed device may not cause both static and dynamic power consumption seriously, and further engineering efforts (enhancing capacitive coupling and/or scaling of the ungated region length) can address this issue. This concern and related brief discussion have been incorporated into the revised manuscript.

We hope that the above discussion on the mechanism of the proposed device will provide a better understanding of our devices and further highlight the significance of our results.

In accordance with the reviewer's comments, in this 3rd revision, Figure 1(a) and some descriptions, including the locations of E_{FS} , E_{FD} , and E_{Fh} (Fermi level of holes), are revised as follows:

Figure 1. (a) Device structure of the WSe₂ I²FET and its energy band profile along the applied bias for each case.

Reviewer comments, fourth round review –

Reviewer #2 (Remarks to the Author):

In the revised manuscript, I am happy to see that the authors improved the band diagram by marking Fermi levels unambiguously, which eliminated potential confusion to readers. Now the main divergence between the authors and me is whether the band bending energy range beyond V_{ds} contributes to impact ionization. 2D materials are generally very defective, compared to Si crystal, the ballisticity of carrier transport in such defective materials cannot be high, which is reason I suspect the authors' explanation. However, I cannot guarantee that such a 'lucky' electron event never happens. I am thinking that maybe it is not a bad idea to report this experiment data first and encourage the broad audience to continue the mechanism exploration. My last comment/request is that the authors should use TCAD or other numerical simulation approach to derive the band diagram instead of a sketch. Using Si channel is fine if 2D channel is not feasible in TCAD. I am requesting this simply because the 'ungated' part in this device is actually gated by a back gate. I suspect the potential drop along the channel happens mainly at 2 local regions, 1) the junction between the part w/ top gate and the part w/o top gate, and 2) drain side, instead of uniformly across the 'ungated' part as depicted by the authors in the existing band diagram sketch. If the simulation results proves my suspicion is correct, then this device is very likely a TFET, instead of an i-MOS. If the simulation results support the authors' current band diagram, I have no issue to recommend to accept this paper.

Response to Reviewer #2

We are grateful for your thorough review of our manuscript. Please find our responses (in blue) to your comments below (in black).

Comments:

In the revised manuscript, I am happy to see that the authors improved the band diagram by marking Fermi levels unambiguously, which eliminated potential confusion to readers. Now the main divergence between the authors and me is whether the band bending energy range beyond V_{ds} contributes to impact ionization. 2D materials are generally very defective, compared to Si crystal, the ballisticity of carrier transport in such defective materials cannot be high, which is reason I suspect the authors' explanation. However, I cannot guarantee that such a 'lucky' electron event never happens. I am thinking that maybe it is not a bad idea to report this experiment data first and encourage the broad audience to continue the mechanism exploration. My last comment/request is that the authors should use TCAD or other numerical simulation approach to derive the band diagram instead of a sketch. Using Si channel is fine if 2D channel is not feasible in TCAD. I am requesting this simply because the 'ungated' part in this device is actually gated by a back gate. I suspect the potential drop along the channel happens mainly at 2 local regions, 1) the junction between the part w/ top gate and the part w/o top gate, and 2) drain side, instead of uniformly across the 'ungated' part as depicted by the authors in the existing band diagram sketch. If the simulation results proves my suspicion is correct, then this device is very likely a TFET, instead of an i-MOS. If the simulation results support the authors' current band diagram, I have no issue to recommend to accept this paper.

Response:

Based on the reviewer's primary comment, we performed band simulation analysis using Synopsys's Sentaurus TCAD (T-2022.03) to verify the band structure of our device. Owing to the lack of 2D materials in the commercial TCAD software, the simulation was performed using Si as a channel material. This may be acceptable because we expect that the simulated band results of Si FETs are similar to those of WSe₂ FETs because the band structures of the two materials are mostly common (i.e., indirect band gap and the gap size).

We performed simulations using a device with the same structure as a WSe₂ I²FET, and the results are presented in Figure R1. Figure R2 depicts TCAD program execution screens for the structure, physical carrier transport model, and material definition.

The simulated results reveal band bending similar to that presented in the original manuscript. The potential energy drops uniformly across the ungated region. Sudden changes in the potential energy near the drain contact region and the homojunction region are not observed.

Based on the simulation results, we can conclude that the gradual band bending observed in the ungated region provides additional energy for carrier multiplication, even though the applied drain voltage is lower than the fundamental limit required for carrier multiplication. Because the simulation results support the schematic band diagrams presented in the manuscript, we hope that our manuscript is now suitable for publication.

- Software: Synopsys's Sentaurus TCAD (T.2022-03)
- Channel material: Silicon
- Band bending at un-gated region = 1.35 eV > $eV_{DS} = 0.88$ eV

Figure R1. (a) TCAD simulated device structure. (b) Energy band diagram obtained using TCAD at $V_{DS} = 0.88$ V.

Figure R2. TCAD program execution screens for the (a) structure, (b) physical carrier transport model, and (c) material definition.

Reviewer comments, fifth round review –

Reviewer #2 (Remarks to the Author):

The authors have addressed my comments. I have no further comments.